# Retracing Hypoxia in Eckernförde Bight (Baltic Sea)

Heiner Dietze[1] and Ulrike Löptien[1]

[1]Institut für Geowissenschaften, CAU Kiel, Ludewig-Meyn-Str. 10, 24118 Kiel, Germany

**Correspondence:** Ulrike Löptien (ulrike.loeptien@ifg-uni.kiel.de)

**Abstract.** An increasing number of dead zoning (hypoxia) has been reported as a consequence of declining levels of dissolved oxygen in coastal oceans all over the globe. Despite substantial efforts a quantitative description of hypoxia up to a level enabling reliable predictions has not been achieved yet for most regions of societal interest. This does also apply to Eckernförde Bight (EB) situated in the Baltic Sea, Germany. The aim of this study is to dissect underlying mechanisms of hypoxia in EB,
to identify key sources of uncertainties and to explore the potential of existing monitoring programs to predict hypoxia - by developing and documenting a workflow that may be applicable to other regions facing similar challenges. Our main tool is an ultra-high spatially resolved general ocean circulation model based on a code framework of proven versatility in that it has been applied to various regional and even global simulations in the past. Our model configuration features a spacial horizontal resolution of $100\,\mathrm{m}$ (unprecedented in the underlying framework which is used in both global and regional applications) and
includes an elementary representation of the biogeochemical dynamics of dissolved oxygen. In addition, we integrate artificial "clocks" that measure the residence time of the water in EB along with timescales of (surface) ventilation. Our approach relies on an ensemble of hind cast model simulations, covering the period from 2000 to 2018, designed to cover a range of poorly known model parameters for vertical background mixing (diffusivity) and local oxygen consumption within EB. Feed-forward artificial neuronal networks are used to identify predictors of hypoxia deep in EB based on data at a monitoring site at the
entrance of EB.

Our results consistently show that the dynamics of low (hypoxic) oxygen concentrations in bottom waters deep inside EB is, to first order, determined by the following antagonistic processes: (1) the inflow of low-oxygenated water from the Kiel Bight (KB) - especially from July to October and (2) the local ventilation of bottom waters by local (within EB) subduction and vertical mixing. Biogeochemical processes that consume oxygen locally, are apparently of minor importance for the development of
hypoxic events. Reverse reasoning suggests that subduction and mixing processes in EB contribute, under certain environmental conditions, to the ventilation of the KB by exporting recently-ventilated waters enriched in oxygen. A detailed analysis of the 2017 fish-kill incident highlights the interplay between westerly winds importing hypoxia from KB and ventilating easterly winds which subduct oxygenated water.

## 1 Introduction

The impact of humans on the Earth System has reached a level of magnitude comparable to natural influences. Among the changes apparently accompanying our way into the Anthropocene are decreasing oxygen concentrations in the global oceans.

This decrease in oxygen is manifesting itself most prominently in coastal regions: in the 1960s only 42 of the so-called "dead zones", that no longer permit the survival of higher animals, have been reported. In 2008 this number has already increased to 400 (Diaz and Rosenberg, 2008). The implications can be substantial, including mass mortality of (commercial) fish, loss of

30 Blue Carbon (associated with seagrass habitat loss), degradation of touristic and recreational assets and release of the potent greenhouse gas $N_2O$ (e.g. Naqvi et al., 2010).

The Baltic Sea in central northern Europe is a prominent example of a coastal region that has been exposed to intermittent dead zoning (i.e. hypoxic events) in the past (Zillén et al., 2008). Apparently hypoxia has increased over time in response to anthropogenic nutrient inputs and ocean warming (Jonsson et al., 1990; Carstensen et al., 2014). Consequently, international

mitigation measures are put into action by the highly industrialized and populated bordering nations (e.g. Helsinki Convention, EU Marine Strategy Framework Directive, Baltic Sea Action Plan) and a discussion of geoengineering options targeted at containing dead zoning has been unbottled (Stigebrandt and Kalen, 2013; Stigebrandt et al., 2015; Liu et al., 2020).

The mechanisms behind the dynamics of oxygen dissolved in seawater are well known: oxygen is produced as a by-product of organic matter production by autotrophs in the sun-lit surface ocean. Organic matter is exported to depth where its rem-

40 ineralization is typically associated with oxygen consumption by bacteria. Air-sea fluxes of oxygen may be in- or outgoing, depending on wether the ocean's surface is over- or undersaturated. Typical surface concentrations of dissolved oxygen are around few hundreds $\mathrm{mmol\,O_2m^{-3}}$, predominantly set by physical solubility as a function of temperature and salinity. Additional complexity is added by the ocean circulation which determines the timescales on which oxygen sources and sinks may accumulate before antagonistic processes set in. This holds especially for the Baltic Sea where sporadic inflows of salty and

45 oxygenated North Sea surface waters replace oxygen-deprived bottom waters of the Baltic Sea (Matthäus, 2006) and where wind-driven upwelling has been identified as a key processes effecting vertical exchange of heat and nutrients (e.g. Lehmann and Myrberg, 2008).

Even though there is consensus regarding the underlying processes, the numerical quantitative simulation of hypoxic conditions remains challenging because it is - essentially - the quest to simulate extremal (low) values, that are determined by

50 the difference of relatively large and uncertain numbers. This introduces high uncertainty to both to the open ocean model applications (e.g. Cocco et al., 2013; Dietze and Löptien, 2013; Löptien and Dietze, 2017) and Baltic Sea model applications (Meier et al., 2011, 2012) which limits their contribution to management or geoengineering decisions of stakeholders. E.g. it has been illustrated in a global model that deficiencies in biogeochemical model components may be compensated by deficiencies in circulation model components (Löptien and Dietze, 2019) thereby obscuring even the sign of the sensitivity of

55 the (global) warming to come. This raises the question if it is actually feasible to reliably (i.e. getting the right answer for the right reason) simulate low-oxygen events in systems such as the Baltic Sea that are (1) infamous for their natural variability (Meier et al., 2021) and (2) subject to antagonistic effects of improved management of water resources and climate change on oxygen concentrations (e.g. Lennartz et al., 2014; Hoppe et al., 2013) - which is notoriously difficult to de-convolve (Naqvi et al., 2010).

The present study steps forward to simulate oxygen dynamics at the exemplary site Eckernförde Bight (EB) which is an appendix to the Kiel Bight (KB) in the German part of the Baltic Sea (Figure 1). The EB site is special in that it hosts the

monitoring station *Boknis Eck* (Figure 2), one of the longest-operated time series stations worldwide (e.g. Lennartz et al., 2014). Consequently, EB is exceptionally well sampled which facilitates the development of numerical models and piloting approaches which may be put to use in other coastal regions threatened by hypoxia (such as other Baltic Sea regions, the East China Sea and Cheasapeake Bay). The overarching aim is to " ... identify critical processes ..." and to " ... provide a supreme dynamic test of knowledge ..." (Flynn, 2005) by simulating hypoxia in EB using a code framework that is proven to be easily applicable globally (e.g. in Dietze et al., 2017), near-globally (e.g. in Dietze et al., 2020) and regionally (e.g. in Dietze et al., 2014). We use an ensemble approach of a suite of regional coupled biogeochemical ocean models targeted at dissecting uncertainties of the biogeochemical module from those of the ocean circulation module. The analyses are aided by integrating artificial tracers measuring residence times - a concept essential to understanding hypoxia (e.g. Fennel and Testa, 2019). Finally, we use an artificial neuronal network (ANN) to identify the critical processes that make the oxygen deficiency deep in the EB predictable - an approach which also gives guidance on the question where uncertainty may lure.

## 2 Methods

MOMBE (**M**odular **O**cean **M**odel **B**ight of **E**ckernförde) is a new configuration of a general ocean circulation model (GCM). The GCM is coupled to a simple representation of biogeochemical processes by introducing an additional passive tracer, that is advected and mixed just like the tracers temperature and salinity but, other than that, controlled by prescribed rates of oxygen production and consumption. Further, we introduce artificial tracers or "clocks" that estimate the residence times and the age (i.e. the time of last contact to the surface) of water parcels. This approach facilitates the dissection between local (i.e. inside EB) and remote (e.g., inflowing hypoxic deep water from the KB) processes that drive the oxygen dynamics. The following subsections describe the GCM, followed by a model evaluation in Section 3. Feed-forward neuronal networks designed to mimic the full-fledged coupled GCM at a station deep inside the Bight, are described in Section 4.4.

### 2.1 Model Configuration

We use the Modular Ocean Model framework MOM4p1, as released by NOAA's Geophysical Fluid Dynamics Laboratory (Griffies, 2009). Model code and configuration are almost identical to those described in Dietze et al. (2014) and Dietze et al. (2020). The few exceptions are listed in the following subsections. Section 2.1.1. describes the model grid, Section 2.1.2 the subgrid parameterizations, and Section 2.1.3 specifies the input data (boundary conditions). Section 2.1.4 documents the representation of sea ice. Section 2.1.5 introduces the implementation of the residence time and age racers. The implementation of the oxygen module is documented in Section 2.1.6.

### 2.1.1 Grid and Bathymetry

The bathymetric data are provided by the Federal Maritime and Hydrographic Agency (BSH, https://www.geoseaportal.de/mapapps/resources/apps/bathymetrie/index.html?lang=de). We use a bilinear scheme to interpolate these onto an Arakawa B model grid (Arakawa and Lamp, 1977). There are $165 \times 103$ grid boxes horizontally, each about $100\,\mathrm{m} \times 100\,\mathrm{m}$ in size

(Figure 2). The total wet area of the model is $119\,\mathrm{km}^2$. The vertical resolution is $1\,\mathrm{m}$, with a total of 31 layers. The average water depth is $11.7\,\mathrm{m}$. The bathymetry was smoothed with a filter similar to the Shapiro filter (Shapiro, 1970). The filter weights are 0.25, 0.5 and 0.25. The filter essentially fills steep holes in the ocean floor which increases numerical stability of the GCM. The filter was successively applied three times, as this has proven (in Dietze and Kriest, 2012; Dietze et al., 2014, 2020) to be a good compromise between unnecessary smoothing on the one hand and numerical instability on the other hand.

### 2.1.2 Subgrid Parameterisations

Even a horizontal resolution as high as $100\,\mathrm{m}$ horizontally and $1\,\mathrm{m}$ vertically fails to explicitly resolve all (turbulent) processes of relevance for transport and mixing of substances in EB. Hence, effects of unresolved small-scale processes have to be parameterized. We use parameterizations and setting identical to those applied by Dietze et al. (2014) in a high-resolution model configuration of the Baltic Sea. An exceptions it the parameter choice for the vertical background diffusivity: Holtermann et al. (2012) estimates from measurements for deep water processes in the central Baltic Sea a vertical diffusivity of $10^{-5}\,m^2\,s^{-1}$ (calculated from the propagation speed of a purposely-deployed dye-like substance). Closer to coast Holtermann et al. (2012) report much higher values. Because mapping this information on conditions in EB is difficult, we decided to test a range of vertical background diffusivities and to assess the respective model perfomances based on available observations. The considered diffusivities are: $5 \times 10^{-5}\,m^2\,s^{-1}$, $1 \times 10^{-4}\,m^2\,s^{-1}$ und $5 \times 10^{-4}\,m^2\,s^{-1}$. This range comprises relatively low diffusivities, which are characteristic for the deep central Baltic Sea, and fairly high values, which are more representative for coastal mixing (as can be expected in the shallow Eckernförde Bight).

### 2.1.3 Boundary Conditions

The atmospheric boundary conditions of our model are set by a reanalysis from the Swedish Meteorological and Hydrological Institute (SMHI). We use the results of the reanalysis framework as a means to interpolate (patchy) observations in time and space. The underlying atmospheric model features a horizontal resolution of $11\,\mathrm{km}$. For the period 2000 to 2015 we use RCA4 (Samuelsson et al., 2015, 2016). RCA4 data is available only until 2015. Hence, for the period 2016 to 2018 we switched to another product: UERRA (regional reanalysis for Europe; https://cds.climate.copernicus.eu/cdsapp#!/dataset/reanalysis-uerra-europe-complete?tab=overview). UERRA is more advanced but does not include "spectral nudging" to the large-scale atmospheric circulation. This detail may allow for unrealistic shifts in the trajectories of low pressure systems. Fortunately, for the time and location under consideration here, a rough comparison with the observations from Kiel lighthouse (in position 54.3344°N,10.1202°E) showed a generally good agreement between reanalysis and direct observations (not shown).

A key element of regional ocean-circulation model configurations are artificial boundary conditions introduced to limit the model domain. Typically, the choice of the extend of the model domain is enforced by computational capabilities rather than by scientific necessity. This can be problematic because boundary conditions are known to introduce spurious effects (e.g. Jensen, 1998; Blayo and Debreu, 2005; Herzfeld et al., 2011). Our choice is pragmatic in that we choose a rigid wall (such as Carton and Chao, 1999; Dietze et al., 2014). In combination with our spacial discretization (Arakawa B Arakawa and Lamp, 1977)

this necessitates a no-slip boundary condition which removes kinetic energy. By this choice, we may underestimate the effect of water entering and leaving the EB. This factor will be considered when analysing the model results.

The water exchange across the rigid wall boundary condition is mimicked by restoring to prescribed temperature, salinity and sea surface height values at the model boundaries only. There is no restoring inside EB and there are no tides because the impact of tides is negligible in the Baltic Sea. For sea surface height we restore to prescribed values taken from an oceanic
reanalysis carried out with MOMBA (Dietze et al., 2014). MOMBA differs from MOMBE in that it covers the entire Baltic Sea with a horizontal resolution of 1 nautical mile while MOMBE introduced here covers the EB only - albeit with much higher resolution (100 m). For the sake of consistency, MOMBA has been integrated for the entire hindcast period 2000-2018 using the atmospheric forcing described above (which differs from Dietze et al., 2014). For temperature, salinity and oxygen we restore MOMBE at its horizontal boundaries with Kiel Bight to interpolated measurements from Station *Boknis Eck* at the
entrance of EB (Lennartz et al., 2014, http://www.bokniseck.de/, http://doi.pangaea.de/10.1594/PANGAEA.855693).

### 2.1.4 Sea Ice

The focus of our investigation are ice-free seasons. We will show in Section 4.1 that the memory of the system under consideration, as given by residence times in Eckernförde Bight, is less than a month. This suggests that sea-ice dynamics are rather irrelevant to the processes and seasons examined here. Even so, for the sake of completeness, we report that our ocean
component is coupled to a dynamical sea ice module, the GFDL Sea Ice Simulator (SIS). SIS uses elastic-viscous-plastic rheology adapted from Hunke and Dukowicz (1997). We use the exact same settings described in Dietze et al. (2020) (which are identical to the settings in Dietze et al. (2014), except for switching to levitating sea ice).

### 2.1.5 Artificial Clocks

In order to facilitate the dissection of local versus remote processes influencing the oceanic oxygen concentrations in EB,
we introduce two artificial tracers or "clocks" to the ocean circulation model (following and approach similar to Dietze et al., 2009). Both clocks behave like dyes in that they are subject to transport processes just like like temperature, salinity and dissolved oxygen. In addition to being transported, the clocks continuously count up time in every grid box. The first clock is reset to zero whenever a water parcel reaches the ocean surface. Thus, it measures the time elapsed since a water parcel had been in contact with the atmosphere. This time is also referred to as the age of the water. The second clock is reset to zero at the
eastern boundaries of the model domain. Thus, it measures the time elapsed since water entered EB. This time is also referred to as the residence time of water in EB.

### 2.1.6 Oxygen

Our dissolved oxygen module is dubbed *EckO$_2$*-module (from **Eck**ernförde **O**$_2$). The module is very similar to the OXYCON approach Bendtsen and Hansen (2013) used also in Lehmann et al. (2014). A schematic representation of *EckO$_2$* is given in
Figure 3. Following Bendtsen and Hansen (2013), the local development over time of dissolved oxygen, $\frac{\partial O_2}{\partial t}$, is defined by:

$$\frac{\partial O_2}{\partial t} + A(O_2) = D(O_2) + S(O_2), \tag{1}$$

where $A$ und $D$ denote the divergence of the three-dimensional advective and diffusive fluxes as calculated by the GCM. $S$ denotes biogeochemical oxygen sources and sinks given by the model parameters *opro* at the sunlit sea surface, by *orewa* at depth below the compensation depth *zco*, and by *orese* in the lowermost wet model grid box. These parameters determine how much oxygen is generated by primary production (*opro*) and how much is consumed at depth (*orewa*) and in the sediment (*orese*). The respective parameter choices are based on literature values listed in Table 1. Following Babenerd (1991) and based on Ærtebjerg et al. (1981) and Jacobsen (1982) we assume that the subsurface oxygen consumption rates are rather uniform throughout KB, EB and up into the Danish Straits. This assumption is necessitated by our lack of direct measurements of consumption rates in EB. $EckO_2$ prescribes climatological monthly mean consumption rates.

Note that our choice of oxygen consumption rates (Table 2) corresponds to a best guess at the higher end of published estimates (Table 1). To this end the simulations including these local sources and sinks of oxygen provide an upper bound on the effects of local biotic processes on local oxygen dynamics in EB. A lower bound is explored by setting local consumption/production to zero.

## 2.2 Observations

We use data from the regular monitoring program of the LLUR. Respective approx. monthly observations of salinity, temperature and oxygen covered the entire hind-cast period at the monitoring station *Buoy 2a* (location marked in Figure 2). Typical surface concentrations of dissolved oxygen are around few hundreds $\mathrm{mmol\,O_2 m^{-3}}$, predominantly set by physical solubility as a function of temperature and salinity (and rather constant atmospheric concentrations). At depth, however, oxygen sinks can accumulate oxygen deficits until critical thresholds for the survival of animal or even plants are undercut. Common denominations for critical thresholds are: *hypoxic, suboxic* and *anoxic* conditions. Their respective values are, however, fuzzy. Here, we follow Gray et al. (2002) and define the threshold values for hypoxia as a concentration of dissolved oxygen of $2\,\mathrm{mg\,O_2\,l^{-1}}$, which corresponds to $\approx 60\,\mathrm{mmol\,O_2\,m^{-3}}$. The relevance of this threshold is that it limits the survival of most fish (Hofmann et al., 2011). In addition we consider a second threshold of $4\,\mathrm{mg\,O_2\,l^{-1}}$ corresponding to $\approx 120\,\mathrm{mmol\,O_2\,m^{-3}}$. This value is used as an indicator for the eutrophication of stratified water bodies (such as EB) by the Baltic Marine Environment Protection Commission (Helsinki Commission - HELCOM, 16th Meeting of the Intersessional Network on Eutrophication Helsinki, Finland, 29.-30. January 2020) and as such of relevance to the stakeholder LLUR.

## 3 Ensemble Generation

Among the challenges in simulating oxygen dynamics is that both biotic parameters (determining oxygen respiration (Section 2.1.6)), and the antagonistic abiotic parameters (that control ventilation with surface water high in oxygen such as e.g. vertical diffusivity (Section 2.1.2)) are uncertain. Our approach is to run an ensemble of simulations encompassing a plausible

range of settings. These settings are listed in Table 2. We compare low, medium and high levels of diffusivity (tagged *HighMix, MedMix, LowMix*, respectively) and, further, simulations which totally neglect local sources and sinks of oxygen (tagged *Rem* for "remote biotic effects only") versus those featuring a best guess of local sources and sinks that is on the higher end of published estimates (cf. Table 1 with Table 2). This section identifies the most realistic simulations which will be considered in the following. The ultimate goal is to chose parameter settings which cover the contemporary uncertainty range.

Figure 4 shows Taylor diagrams which compare simulated and observed temperature, salinity and oxygen. The simulations with high diffusivity (*HiMix* and *HiMixRem*) feature the lowest performance in reproducing the observed variability of temperature, salinity and oxygen. This is consistent with an assessment of simulated velocities by Marlow (2020). We thus discard these simulations from the analysis. The more realistic simulations *LoMix* and *HiMix* are very similar - irrespective of wether we account for local sources and sinks of oxygen or not. We conclude (from Figure 4) that the lower values for the diffusivity are more realistic and that local sources and sinks of oxygen are apparently of minor importance within EB.

Figure 5 shows simulated and observed oxygen concentrations at the bottom of the monitoring station *Buoy 2a* for the years 2000 - 2015. Shown are the respective months April to October. November to March are omitted because these months feature high concentrations of dissolved concentrations beyond our scope of interest. The overall impression is that the model retraces the dynamics of temperature, salinity and oxygen reasonably well. Figure 6 provides a more quantitative estimate of the fidelity in reproducing hypoxic events (as defined by the $120 \, \mathrm{mmol} \, O_2 \, \mathrm{m}^{-3}$ introduced in Section 1) at the monitoring station *Buoy 2a*. It shows sensitivity and specificity achieved with the simulations *LoMix* and *MedMix* that account for local sources and sinks of oxygen: *LoMix* typically simulates $\approx 70\%$ true positives and $\approx 10\%$ false positives. *MedMix*, in comparison, simulates only several % false positives but fails to identify every third event ( i.e., $\approx 70\%$ true positives).

## 4 Results

We start with exploring the simulated residence and ventilation timescales (Section 4.1) for the simulations *LoMix* and *MedMix*. This provides a base for understanding the dynamics behind our hind cast, presented in Section 4.2. A complementary case study of the intense hypoxic event 2017 is presented in Section 4.3. Section 4.4 describes the application of artificial intelligence for feature selection and extraction of the predictive capability of monitoring data at Station *Boknis Eck* at the entrance of EB to forecast hypoxia within EB at the monitoring station *Buoy 2a*.

### 4.1 Residence and Ventilation Times

The estimates of *residence* and *ventilation times* are calculated with "artificial clocks", as described in Section 2.1.5. Both model versions *LoMix* and *MedMix* show similar results: the water with the longest residence time is found at the end of EB in the interior close to the city Eckernförde (Figure 7). Typical values are of the order of one month for both exemplary months, August and October. Overall, *MedMix* shows lower values than *LoMix* indicating that vertical diffusive processes promote the horizontal exchange of water between EB and KB. This makes sense because the longest residence times can be found at the surface (Figure 8), suggesting that, on average, water enters the Bight at depth and leaves the Bight at the surface. A stronger

vertical diffusivity is then associated with an accelerated rate of surface water renewal by deep water with shorter residence times.

The distribution of ventilation times or age is similar to that of residence times in that the highest values are generally found within the Bight towards Eckernförde (Figure 9). The horizontal gradient is more pronounced in the simulation with lower mixing, while higher prescribed vertical background mixing equalizes the effective ventilation processes horizontally. In terms of vertical distribution age has, in contrast to the residence time, high values at depth and low at the surface - where it is reset to zero (Figure 10).

In summary, we find that residence times and age are of similar magnitude. This suggests that the first order control of processes that determine oxygen concentrations in EB is an antagonistic interplay of inflowing water (generally low in oxygen) and the local aeration by vertical exchange with oxygenated surface waters. Biogeochemical processes in the interior of EB are apparently of minor importance for the oxygen dynamics within EB.

## 4.2    The Typical Seasonal Cycle inside EB

Figure 5 shows a comparison between the observed and simulated temporal evolution of dissolved oxygen concentrations at the bottom of the monitoring station *Buoy 2*. Most prominent is a pronounced seasonal cycle. The generic explanation for such seasonal cycles in such latitudes is that temperatures and biomass production in the surface waters ramps up in spring - driven by enhanced levels of photosynthetically available radiation (note, however that there is an ongoing discussion on this issue Behrenfeld, 2010; Arteaga et al, 2020; Smetacek, 1985). The biomass eventually sinks to depth where it degrades and

issues oxygen consumption. Later in the season, the water column stratifies and the surface layer heats up, effectively creating a barrier to the exchange of bottom water (deprived in oxygen) and the oxygenated surface waters. As autumn approaches, the surface ocean cools again and weakens the stratified barrier to vertical mixing. This facilitates the wind-driven mixing events that come along with more unstable autumn weather. In winter, convective mixing homogenizes the entire (rather shallow) water column vertically (e.g., Fennel and Testa, 2019; Petenati, 2017). Apparently the model captures this dynamic well, i.e.,

the ensemble mean of *LoMix* and *MedMix* features a high visual correspondence between the respective curves in Fig. 5 (see Figure 4 for more quantitative estimate).

     Based on the hind cast simulation from 2000-2015 hypoxic events at Station *Buoy 2a* are most common in August and October with a local minimum of occurrences in September (Figure 11). This is inconsistent with the generic explanation outlined above, where a period of ever decreasing levels of dissolved oxygen ends in autumn when increasing winds and a

245 pronounced air-sea heat transfer promotes net ventilation. So why do hypoxic conditions deep in EB at Station *Buoy 2a* become more frequent after the September setback, despite increasing winds and decreasing thermal stratification? The histograms of bottom oxygen concentrations observed at Station *Boknis Eck*, situated at the entrance to EB (and used to prescribe the conditions of water flowing into EB in the model), suggest: particularly low oxygen concentrations are more frequent in October than in August (Figure 12). Hence, water entering EB from KB in October are more likely to "import" hypoxia.

Note that these considerations are inline with simulations *LowMixrem/LowMix* and *MedMixrem/MedMix* each of which pairs showing in Figure 4 very little effect of local oxygen consumption within EB - even though: (1) the respective biotic local

oxygen consumptions are chose to represent the upper limit of published estimates and (2) the water exchange with KB is hampered by a rigid wall boundary condition.

We conclude: the typical oxygen deficit in late summer is imported along with water from the KB, rather than being produced locally in EB. The following Section 4.3 will elucidate the underlying succession of events by means of a detailed case study.

## 4.3 Hypoxic Event 2017

In fall 2017 a particularly pronounced hypoxic event occurred and led to a mass fish kill incidence. In the following, we analyze this event in the MOMBE *LoMix* simulation.

Figure 13 shows a sequence of snapshots of simulated hypoxia in EB, starting August 20th and ending at peak conditions on September 10th. Over the course of these several weeks, EB looses oxygen and hypoxic waters apparently enter the Bight at the bottom from the east and moves upwards. The notion of "imported" hypoxic conditions is backed by the Hovmoeller Diagrams of simulated age and residence times at the monitoring station *Buoy 2a* in Figure 14: during the buildup of the hypoxic event in EB, the residence time features a local minimum deep inside EB. This suggests the prevalence of water masses "recently-imported" from KB (Figure 14 b). Simultaneously, the age features a maximum (Figure 14 a), indicating that the "recently-imported" hypoxic waters are well-shielded from ventilation by oxygenated surface waters. Further evidence is provided by Figure 15, showing that the oxygen decline in EB is contemporaneous with winds blowing out of the Bight. These winds drive an overturning circulation, shown in Figure 16, with surface waters being pushed out of the Bight and bottom waters, for continuity reasons, being sucked into the Bight at depth. Consequently, we find in Figure 15 that the oxygen decline at the entrance of the Bight (at Station *Boknis Eck*) occurs earlier than the oxygen decline inside the Bight (at Station *Buoy 2a*) - just as expected in a system where water enters the Bight at the bottom.

During the relaxation phase, that terminates the 2017 hypoxic event, the processes are reversed: Figure 17 shows that the winds are blowing consistently into the Bight for more than a week. Consequently, water is pushed into the Bight at the surface, having nowhere to go. Some of the well-oxygenated surface water is subducted to depth and subsequently leaves EB at depth. Just as expected, the increase in oxygen at the monitoring station *Buoy 2a* inside the Bight occurs earlier than the corresponding oxygen increase at the entrance Station *Boknis Eck*). The oxygen levels at *Boknis Eck* now lag behind *Buoy 2a* by approximately one week.

In summary, we identified a governing mechanism by which EB is - depending on wind direction - either: (1) impacted by imported low oxygenated waters from KB or (2) being flushed by oxygenated surface water, that is subducted to depth in the interior of EB and is exported at depth to KB - whereby EB is effectively ventilating KB.

Open question, however, remain. Of particular interest is the questions why some years are hit especially hard by hypoxia and wether such events are predictable days or weeks in advance. Such predictions may, e.g., allow for netting and landing of doomed fish. The following section applies Artificial Intelligence (AI) to pursue these questions.

## 4.4 AI-based feature selection and time series prediction

The following section explores the statistical relations between the simulated time series at Station *Buoy 2a* deep in the Bight and *Boknis Eck* at the entrance of the Bight. The major aims are: (1) To gain further mechanistic insight. (2) To develop a surrogate models for the stakeholder that may be implemented on off-the-shelf desktop computers, smart phones or even on very low cost ($< 10$,- Euros) embedded devices rather than necessitating access to a super-computing facility (as is the case with the full-fletched coupled model). This section is motivated by recent and encouraging success in emulating general circulation models (e.g Castruccio et al., 2014), ecosystem models (e.g. Fer et al., 2018), the tremendous success in machine learning and data-driven methods in fluid dynamics (as summarized e.g. by Brunton et al., 2020) and the sneaking suspicion that " ... the most pressing scientific and engineering problems of the modern era are not amenable to empirical models or deviations of first principles ..." (Brunton et al., 2020b).

In the following, we describe the application of shallow and deep feed-forward artificial neuronal networks (ANNs) to forecast bottom oxygen concentrations deep inside EB at the monitoring station *Buoy 2a* two weeks in advance from the atmospheric conditions and the regularly sampled monitoring station *Boknis Eck* at the entrance of the Bight. The forecast range is chosen as a compromise between the time needed for mitigation measures (e.g. by netting and landing of doomed fish) and forecast accuracy which typically degrades with forecasting range. During the course of this exercise we will use different combinations of predictors (or input data) and test their impact on the forecast skill - a processes also referred to as capacity estimation and feature selection (e.g., Sbalzarini et al., 2002). Note, however, that a comprehensive analysis of time series forecasting, which must include traditional statistical approaches in addition to machine learning methods (Makridakis et al., 2018), is beyond the scope of this manuscript.

### 4.4.1 Capacity Estimation and Feature Selection

For training the ANNs, we draw our training (80%) and validation data (20%) randomly from the 2000 to 2016 model hind cast. We hand-design features (input data) and test their respective capacity to forecast bottom oxygen concentrations at Station *Buoy 2a* (target data). Hand-designed features are "... two edged swords" (e.g. Reichstein et al., 2019): they can be seen as an advantage because they give us control of the explanatory drivers which may be used to promote system understanding. On the other hand, hand-designed features are typically suboptimal. To this end our results here provide a lower bound on the potential of ANNs for the task at hand.

The ANN is trained using the Levenberg-Marquardt algorithm (Marquardt, 1963) applied to neuronal network training following (Hagan and Menhaj, 1994) and (Hagan et al., 1996). Each training is repeated 30 times, each of which may yield (slightly) differing results because: depending on the (random) initialization of weights, the algorithm may terminate in potentially differing local optima of the cost function. As cost-function we choose mean-squared errors (calculated from MOMBE output and the ANN prediction designed to mimic the MOMBE output). Figure 18 shows respective cost as errors relative to a naive biweekly persistency forecast based on bottom oxygen concentrations at the monitoring station *Boknis Eck*: apparently

the ANN's performance converges at 45% relative to the persistency forecast. Defining this as the Pareto Frontier suggests a Pareto Optimal of 56% - which corresponds to one or two nodes. The idea of opting for a rather parsimonious two-node model that scores 80% of the Pareto Frontier rather than 100% is to reduce the risk of overfitting which may hinder generalization. Further, parsimonious models are easier to interpret than their complex counterpart such that their robustness is easier to assess. This is especially important because we have no straightforward way to extract human semantics from the "rules" the neuronal network learned during the optimization process that related our input features to the target bottom oxygen concentrations at Station *Buoy 2a*.

We start with a shallow (one input, one hidden and one output layer) ANN utilizing the full vertical profiles of temperature, salinity and oxygen along with a biweekly wind forecast totaling at 106 input features (given by the three 1-m resolution vertical profiles of temperature, salinity and oxygen down to 26 m depth and the 14-daily forecasts of zonal and meridional winds each). This setup is based on an optimistic estimate of the number of features available to stakeholders. Specifically, we assume to have access to a correct biweekly wind forecast along with one full vertical profile of each temperature, salinity and oxygen at the monitoring station *Boknis Eck* located at the entrance of EB (i.e., the 106 features introduced above).

Figure 18 suggests that the Pareto Frontier is at 45% corresponding to a 55% reduction in error relative to the persistence model. 80% of this yields a Pareto Optimal of 56%. This corresponds to one or two nodes. Additional tests with deeper ANN's featuring up to 10 hidden layers with two nodes were unsuccessful in that respective errors were always higher than 50%. We conclude that a simple two node shallow ANN features already a reasonable performance and two input features, of the 106 tested, may suffice to capture the main effects.

Table 3 summarizes our effort to identify the most predictive features by backward elimination (e.g. Dietterich, 2002). Using combinations of only 15 features comprised of biweekly zonal windspeed and the bottom values of either temperature, salinity or oxygen yielded a moderate degrade in performance of only 10% (Table 3 entries 2 to 4). Pushing further we identified a combination of two features only that are, on the one hand, within this 10% degradation and, on the other hand, especially easy to measure for stakeholders: surface and bottom temperature at Station *Boknis Eck*. Counter to intuition adding wind forecasts does not improve the ANNs fidelity (compare entries 5 and 6 in Table 3). Even so, the ANN fits the training and validation data remarkably well (Figure 19). We conclude that the ANN's biweekly forecast exploits links other than those being direct consequences of the wind driven inflow versus ventilation mechanism identified in Section 4.3. Section 4.4.2 puts this exploitation to the test using independent test (model) data.

### 4.4.2 ANN Generalization

This section discusses the fidelity of the two-node ANN using simulated bottom and surface temperature identified in Section 4.4.2 as being parsimonious but - even though - yielding reasonable results compared to more complex architectures, such as deeper nets using more nodes and input data. Here, we use independent test data covering the years 2016 to 2018 of our hindcast simulation. This data has neither been used in training nor during validation so far. To rate the forecast it is compared to the "persistence model", which assumes that the oxygen concentrations at station *Boknis Eck* appear two weeks later at station *Buoy 2a* (green line in Figure 20). The first striking impression of the close-ups in Figure 20 is that all years feature a

similar seasonal decline in bottom oxygen in autumn and this decline generally closely resembles the oxygen decline in *Boknis Eck* two weeks in advance. Large interannual differences, however, occur in the onset of the trend reversal. This "return-point" in time is not captured well by the persistency model. These results are consistent with our results in Section 4.3 showing that the decline is driven by the import of low-oxygenated waters from KB. Ventilation, however, takes place in the interior of the Bight and its signal reaches Station *Boknis Eck* at the entrance afterwards - such that we indeed expect no predictive power of the persistency model under these circumstances. To this end, our ANN clearly outperforms the persistency model in that it predicts an earlier and more realistic recovery of oxygen values during end of summer / beginning of autumn - despite the ANN also exclusively relying on data at the entrance at Station *Boknis Eck*.

The ANN essentially and successfully links information regarding season ("derived" from sea surface temperature) and stratification ("derived" from the temperature difference between surface and depth) at the entrance of the Bight with oxygen concentration in the interior of the Bight - without utilizing information on winds. This clearly emphasizes the role of stratification in putting an end to hypoxic events: EB is in the latitudes of prevailing westerlies with "prevailing" entailing that the local winds shift back and forth as the weather systems travel east. Any of these wind shifts from westerly to easterly may end an hypoxic event in EB - if the stratification is weak enough (and winds are strong enough) such that oxygenated surface water can be pushed to depth. In a nutshell: if the stratification has sufficiently weakened you know that that the next wind shift will subduct oxygenated water thereby ending the hypoxic event.

In summary, the ANN features a remarkable performance given that it simply relies on two temperature measurement at the entrance of the Bight. This performance is owed to the importance of stratification in setting the length of hypoxic events: Eroding stratification preconditions the wind-driven downwelling or subduction of oxygenated surface waters which ends hypoxic events. Given that the EM is positioned in the prevailing westerlies the winds regularly change to easterlies - but this does only drive substantial oxygenation (replacement) of bottom waters if the stratification is weak enough to be penetrated. Hence, the high explanatory power of surface and bottom temperature at the entrance of EB to predict the dynamics of hypoxia deep in EB.

## 5 Discussion

Oxygen concentrations are controlled by the antagonistic interplay of respiration and ventilation processes - both of which may respond antagonistically to climate change and improved management of water resources (e.g. Lennartz et al., 2014; Hoppe et al., 2013).

Our model-based analysis suggest that the variability in the occurrence of hypoxic conditions in EB is correlated with the a high variability in wind-driven ventilation rather than with a high variability in local respiration. This result is in agreement with Ærtebjerg et al. (2003), who examined the massive 2002 (one of the worst ever documented) oxygen deficit event that encompassed the Kattegat, the Belt Sea and the Western Baltic Sea. Back then, Ærtebjerg et al. (2003) found no evidence for anomalous respiration patterns i.e. metrics like anthropogenic phosphate loads and the evolution of the phytoplankton spring

bloom appeared to have stayed - in contrast to the oxygen concentration - within typical bounds. This, in turn, highlighted the importance of the variability of ventilation in shaping hypoxic events.

In our model frameworks we distinguish between two types of ventilation: for one, vertical mixing driven by isotropic turbulence and composed of a parameterization of constant background mixing complemented by a surface mixed layer model that mimics the effect of convection, shear-instability and wind-induced turbulence (more specifically we use the KPP scheme of Large et al., 1994). Vertical mixing is difficult to constrain in models because direct observations of turbulence are rare and additional complexity arises from numerical subtleties in models (e.g. Burchard et al, 2008). That said, we use the fidelity of

simulated temperatures as a proxy for the realism of mixing rates: our simulations *LoMix* and *MedMix* featuring a vertical diffusivity of $5 \times 10^{-5}\,\mathrm{m}^2\,\mathrm{s}^{-1}$ and $10^{-4}\,\mathrm{m}^2\,\mathrm{s}^{-1}$ both fit the observations inside the Bight reasonably well. The respective correlation coefficients are $\approx 0.9$ at a simulated standard deviation scoring $\approx 90\%$ of the observed (Figure 4). This is roughly consistent with an estimate inferred from the rate of spreading of a deliberately released substance from Holtermann et al. (2012) who report a basin-scale Baltic Sea vertical diffusivity of the order of $10^{-5}\,\mathrm{m}^2\,\mathrm{s}^{-1}$ with dramatically increasing values

in proximity to the coast.

The other type of ventilation that is of relevance in our coupled ocean-circulation biogeochemical model is the explicitly resolved (i.e., not parameterized) wind-driven overturning circulation in EB (Figure 16). There is consensus that wind-driven vertical circulation is a key mechanism in the Baltic Sea (e.g. Lehmann and Myrberg, 2008) including EB (Karstensen et al., 2014). Wind-driven vertical circulation is associated with upwelling only most of the times - simply because upwelled waters are typically cold and nutrient-rich which may be easily traced by satellites resolving cold filaments and spawning phytoplank-

ton blooms both in space and time. Less prominent is the effect of wind-induced downwelling. Driven by a convergence of surface water such events typically do not manifest themselves in surface properties and, consequently, are rarely discussed. A closer look into our simulated seasonal cycles of the years 2016, 2017 and 2018 (Figure 20), however, showcases the importance of (often-ignored) wind-driven downwelling in controlling hypoxia in EB: we find that the minimum oxygen concentration is

mainly set by the timing of the first overturning event in late summer / beginning of autumn when winds push surface waters into the Bight where it is subducted, overcomes the vertical stratification and replaces deoxygenated bottom waters with recently oxygenated surface waters. This explicitly resolved overturning cycle expands over the whole Bight and apparently exports oxygenated bottom waters thereby ventilating KB. Given the reasonable representation of the seasonal cycles during the 2000 to 2016 period (Figure 5) we conclude that our coupled ocean-circulation biogeochemical model resolves the major

processes at play - although at a high computational cost.

Further mechanistic insight resulted from an exploration of the relations between simulated time series at Station *Buoy 2a* in the interior of EB and biweekly lagged series at station *Boknis Eck* at the entrance of the Bight using an ANN: Counter to our intuition, an ANN fed with information on stratification (i.e. bottom and surface temperature whose difference is a measure of stratification) at the entrance of the Bight and season (i.e. surface temperature which is strongly correlated to season) only,

performs surprisingly well without access to wind forecasts - even though the major mechanism behind the oxygen variability is wind-driven. This highlights the importance of the preconditioning that has to precede a ventilating overturning event: In EB, deoxygenation continues almost monotonically until destabilizing buoyancy fluxes have eroded the stability of the water

column to a point where the next shift to easterly wind can replace the denser bottom waters with lighter surface waters. Because synoptic weather systems and associated wind directions have a lifetime of the order of a week in EB, forecasts based on state of preconditioning are, on average, accurate within a week.

So although the wind-driven upwelling and, especially, the downwelling (wich traditionally is not so much in focus because its effects are not as evident at the easy-to-observe surface) is the key process driving oxygen dynamics, we identified the stratification to be the ultimate gatekeeper for determining the length and severity of seasonal hypoxia in EB. This results relates hypoxia in EB directly with climate change because increased oceanic stratification is driven by a warming atmosphere.

But caveats remain. Among those is the influence of the waste water treating facility Kiel Bülk. Kiel Bülk serves 310.000 citizens and discharges $19 \times 10^6\,\mathrm{m}^3$ treated sewage per year to the sea close to our model boundary. Our model calculations do not account for this because we lack respective data on sewage composition. The following back-of-the-envelope calculation based on published data covering an extreme event puts the potential influence of Kiel Bülk into perspective: Haustein (2002) documents a discharge corresponding to $24.4\,\mathrm{tons}$ of COD (chemical oxygen demand) for the extreme heavy rain event of July 18, 2002. This corresponds to $7.6\,10^5\,\mathrm{mol}\,O_2$. Our model domain covers roughly a wet area of $120\,\mathrm{km}^2$ with an average depth of $11.7\,\mathrm{m}$, corresponding to a volume of $1.4\,10^9\,\mathrm{m}^3$. Hence, assuming that currents swept the entire discharge of July 18th into EB where it spread out homogeneously yields a reduction of only $1\mathrm{mmol}\,O_2\,\mathrm{m}^{-3}$. This is negligible - to the extent that the assumption of instantaneous homogeneous distribution over the entire Bight holds.

Another issue that surfaced in the review process is boundary conditions. Our model domain ends east of Middelgrund with a rigid wall which introduces spurious effects. Note that this applies to all boundary conditions (e.g. Blayo and Debreu, 2005; Herzfeld et al., 2011; Jensen, 1998) because there is, inevitably, a price to be payed for the benefit of not having to resolve the entire ocean (and pay the associated computational cost). In our case the Arakawa B model grid (Arakawa and Lamp, 1977) discretization necessitates a no-slip boundary condition effectively taking kinetic energy out of the system. Even so we find that the thereby (spuriously) damped circulation is the key process in that it, on the one hand, imports hypoxia into EB and, on the other hand, subducts oxygenated surface waters. We argue that this result is robust towards the choice of boundary condition because open boundary conditions (as opposed to the rigid wall we use here) are prone to allow an even more vivid circulation.

## 6 Conclusions

Oxygen concentrations are controlled by the antagonistic interplay of respiration and ventilation processes - both of which may respond antagonistically to climate change and improved management of water resources (e.g. Lennartz et al., 2014; Hoppe et al., 2013). The quantitative estimation of respective sensitivities is painstaking also because of the systems intrinsic large natural variability (e.g. Meier et al., 2021). But it is without alternative if well-intentioned policy is to effectively combat coastal hypoxia in a warming world featuring already more than 60 patents on artificial downwelling techniques Liu et al. (2020).

We set out to dissect the mechanisms driving hypoxic events and associated fish-kills in EB and to identifiy the major sources for uncertainties in the underlying model. We developed the high-resolution coupled ocean-circulation biogeochemical model MOMBE and integrated an ensemble of hind cast simulation covering the years 2000 to 2018. Our analysis based on

simulated and observed oxygen, temperature and salinity along with artificial model tracers quantifying residence times and local ventilation (*ideal age*) revealed the two major and antagonistic processes determining oxygen variability in EB: (1) The oxygen deficit in EB which builds up every summer is imported from KB. The prevailing westerlies push surface water out of the Bight. Its replacement enters the Bight at depth which, in summer, taps into the oxygen depleted deep(er) KB. Local

oxygen consumption in EB plays a minor role in shaping hypoxic events. (2) Intermittent easterly winds subduct oxygenated surface water at the end of the Bight - once the vertical stratification has been sufficiently degraded in late summer / beginning of August. The subducted water ventilates the entire EB and, as it is exported to KB, contributes to ventilating KB.

Further, we explored the predictability of hypoxia in the interior of EB (at Station *Buoy 2a*) based on data from the entrance (at Station *Boknis Eck*). The rationale was to identify main controlling mechanisms and to develop a computationally cheap

forecasting tool for stakeholder. Successful experiments with an Artificial Neuronal Network, trained with data from the coupled MOMBE model revealed in a backward elimination exercise that surface and bottom temperature on their own (taken at a monitoring station at the entrance of EB) provide enough information for a reasonable biweekly forecast of bottom oxygen concentrations deep in EB. This finding traces the severity of hypoxia in late summer as being a consequence of a wind-induced subduction of surface water that is delayed (or advanced) by the state of stratification. More specifically we identified a system

where the severity of seasonal hypoxia is clearly controlled by wind-induced downwelling gatekept by stratification.

Our approach to simulate local hypoxia with high-resolution models and then identify the key processes by ways of machine learning is versatile in that it may easily be applicable to other regions affected by hypoxic conditions. Given that there are already more than 60 artificial downwelling techniques patented (Liu et al., 2020) - which may or may not be put to work to contain coastal hypoxia in our warming future - we rank a more comprehensive quantitative system understanding of local

hypoxia all over the world among pressing societal questions.

*Code and data availability.* The circulation model code MOM4p1 is distributed by NOAA's Geophysical Fluid Dynamics Laboratory (http://www.gfdl.noaa.gov/fms). We use the original code without applying any changes to it. Meridional sections and bottom values of simulated oxygen concentrations, temperature, salinity, residence time and age have been visualized for the hind cast period 2000-2018 for the stakeholder. They are archived under https://doi.org/10.5281/zenodo.4271941 and accessible via https://doi.org/10.5281/zenodo.4271941.

The Boknis Eck Time-Series Station is run by the Chemical Oceanography Research Unit of the GEOMAR Helmholtz Centre for Ocean Research Kiel. The data from Boknis Eck are available from www.bokniseck.de/database-access.

*Author contributions.* H. Dietze and U. Löptien have been equally involved in setting up and running the model configurations. Both authors contributed to the interpretation of model results, to outlining and writing of the paper in equal shares.

*Competing interests.* The authors declare that they have no conflict of interest.

*Acknowledgements.* We acknowledge support by Birgit Schneider. This work is part of an collaborative effort between the Christian-Albrechts-Universität zu Kiel and the Landesamt für Landwirtschaft, Umwelt und ländliche Räume (LLUR) titled *Frühwarnsystem Upwelling (FRAM), Vergabenummer 0608.451812*. We are grateful to the MOM community for sharing code and expertise. The key figure is based on symbols distributed by https://ian.umces.edu/symbols/, courtesy of the Integration and Application Network, University of Maryland Center for Environmental Science. We are grateful to expertise conveyed to us by the RedMod project (https://redmod-project.de/).

Specifically, we acknowledge support by, and discussions with Corinna Schrum and Udo von Toussaint. The Bachelor student Jonas Marlow supported initial model evaluation. Ute Hecht from the maritime meteorology department at the GEOMAR Helmholtz Centre for Ocean Research Kiel provided the weather data from Kiel Lighthouse. LLUR provided data from monitoring Station Buoy 2a. The Chemical Oceanography Research Unit of GEOMAR provided Boknis Eck data. We acknowledge discussions with Rolf Karez, Ivo Bobsien, Britta Munkes and all participants of the 2018 *Freundeskreis Eckernförde* meeting. The constructive discussion with anonymous reviewers medi-

ated by our editor T. Treude has helped to substantially improve an earlier version of this manuscript. Thank you all for your interest, time and effort!

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

**Table 1.** Estimates of oxygen consumption and production converted to respective model parameters of the EckO$_2$ module. Conversions may include devision by the average water depth and area of Eckernförde Bight (see Section 2.1.1), a O$_2$:C ratio of 1.1 and a C:P ratio of 106.

| Reference | Description | *opro* $\left[\frac{\text{mmol O}_2}{\text{m}^2\,\text{day}}\right]$ | *orewa* $\left[\frac{\text{mmol O}_2}{\text{m}^3\,\text{day}}\right]$ | *orese* $\left[\frac{\text{mmol O}_2}{\text{m}^2\,\text{day}}\right]$ |
|---|---|---|---|---|
| Babenerd (1991) | In-situ measurements during summer stratification 1985 & 1986 at the monitoring station *Boknis Eck* | | 3.75 | |
| Bendtsen and Hansen (2013) | Prescribed parameters in a model of the Baltic Sea-North Sea transition which yielded a good fit to observed oxygen concentrations | 2.75 | 0.36 | 3.1 |
| Rahm (1987) | Budget calculations for the Baltic Proper | | 0.26 | |
| Noffke et al. (2016) | In-situ measurements with a lander in the Eastern Gotland Basin | | | 5.8 - 20.8 |
| Pers and Rahm (2000) | Budget calculations for the Baltic Proper | | 1.1 - 2.4 | |
| Smetacek (1980, 1985) | In-situ measurements in the western Kiel Bight with detritus traps in June (assuming negligible fraction of permanent burial) | | | 1.6 |
| Smetacek (1980, 1985) | In-situ measurements in the western Kiel Bight with detritus traps in August (assuming negligible fraction of permanent burial) | | | 6.3 |
| Haustein (2002) | Average (dry days) oxygen consumption equivalent of Kiel Bülk sewage effluent, distributed evenly over Eckernförde Bight | | 0.04 | |
| Haustein (2002) | Episodic, extreme discharge event during 18th and 19th July 2002 of the Kiel Bülk sewage plant, converted into oxygen consumption equivalent distributed evenly over Eckernförde Bight | | 0.36 | |
| Nausch et al. (2011) | Average Kiel Bülk sewage phosphorous effluent, converted into oxygen consumption assuming that it fuels organic matter production that is remineralized in Eckernförde Bight | | 0.03 | |
| Nausch et al. (2011) | Phosphorous loads of rivulet Schwentine that drains into Kiel Bight, converted into oxygen consumption assuming that it fuels organic matter production that is entirely remineralized at depth in Eckernförde Bight | | 0.18 | |

**Table 2.** List of model parameter settings for the EckO$_2$-module and diffusive background mixing in MOMBE. $\kappa_v$ refers to vertical background mixing (diffusivity). *opro*, *orewa* and *orese* refer to monthly (one value per month starting with the January value) oxygen production, water column oxygen respiration and oxygen consumption by the sediment, respectively (cf. Figure 3). Values for *orewa* and *orese* are derived from the published estimates listed in Table 1. *opro* is calculated as residual assuming instant equilibration of sedimentary fluxes.

| tag | description | $\kappa_v$ $m^2\,s^{-1}$ | *opro* [mmol O$_2$ m$^{-2}$ day$^{-1}$] | *orewa* [mmol O$_2$ m$^{-3}$ day$^{-3}$] | *orese* [mmol$_2$ m$^{-2}$ day$^{-1}$] |
|---|---|---|---|---|---|
| *LoMix* | Low vertical background mixing of momentum and tracers. Local oxygen consumption/production rates at the upper limit of published estimates. | $5 \times 10^{-5}$ | 48 47 47 46 46 45 48 50 50 49 48 48 | 3.8 3.8 3.8 3.8 3.8 3.8 3.8 3.8 | 4 3.5 3 2.5 2.1 1.6 3.95 6.3 5.8 5.4 4.9 4.4 |
| *LoMixRem* | Low vertical background mixing of momentum and tracers. No local oxygen consumption/production. | $5 \times 10^{-5}$ | 0 0 0 0 0 0 0 0 0 0 0 0 | 0 0 0 0 0 0 0 0 0 0 0 0 | 0 0 0 0 0 0 0 0 0 0 0 0 |
| *MedMix* | Medium vertical background mixing of momentum and tracers. Local oxygen consumption/production rates at the upper limit of published estimates. | $1 \times 10^{-4}$ | 48 47 47 46 46 45 48 50 50 49 48 48 | 3.8 3.8 3.8 3.8 3.8 3.8 3.8 3.8 | 4 3.5 3 2.5 2.1 1.6 3.95 6.3 5.8 5.4 4.9 4.4 |
| *MedMixRem* | Medium vertical background mixing of momentum and tracers. No local oxygen consumption/production. | $1 \times 10^{-4}$ | 0 0 0 0 0 0 0 0 0 0 0 0 | 0 0 0 0 0 0 0 0 0 0 0 0 | 0 0 0 0 0 0 0 0 0 0 0 0 |
| *HiMix* | High vertical background mixing of momentum and tracers. Local oxygen consumption/production rates at the upper limit of published estimates. | $5 \times 10^{-4}$ | 48 47 47 46 46 45 48 50 50 49 48 48 | 3.8 3.8 3.8 3.8 3.8 3.8 3.8 3.8 | 4 3.5 3 2.5 2.1 1.6 3.95 6.3 5.8 5.4 4.9 4.4 |
| *HiMixRem* | High vertical background mixing of momentum and tracers. No local oxygen consumption/production. | $5 \times 10^{-4}$ | 0 0 0 0 0 0 0 0 0 0 0 0 | 0 0 0 0 0 0 0 0 0 0 0 0 | 0 0 0 0 0 0 0 0 0 0 0 0 |

**Table 3.** Capacity estimation of input features. This table relates the fidelity of biweekly walk-forward ANN forecast of bottom oxygen concentrations at the monitoring station *Buoy 2a* with data from Station *Boknis Eck* fed to the ANN. The average of windspeed squared refers to respective biweekly forecast of zonal winds. The error is the RMS deviation between the (computationally cheap) ANN projection and simulated (computationally expensive; full-fledged coupled biogeochemical ocean circulation model) bottom oxygen concentrations at Buoy 2a relative to the respective RMS of the persistence model (which naively assumes that *Boknis Eck* bottom oxygen concentrations will persist for 14 days at Buoy 2a.

| Input Features | Error [%] |
|---|---|
| average of zonal and meridional windspeed squared, full vertical profiles (26 depth levels) of $O_2$, temperature and salinity | 54 |
| average of zonal windspeed squared, bottom $O_2$ | 64 |
| average of zonal windspeed squared, bottom salinity | 65 |
| average of zonal windspeed squared, bottom temperature | 62 |
| average of zonal windspeed squared, surface and bottom temperature | 58 |
| surface and bottom temperature | 58 |

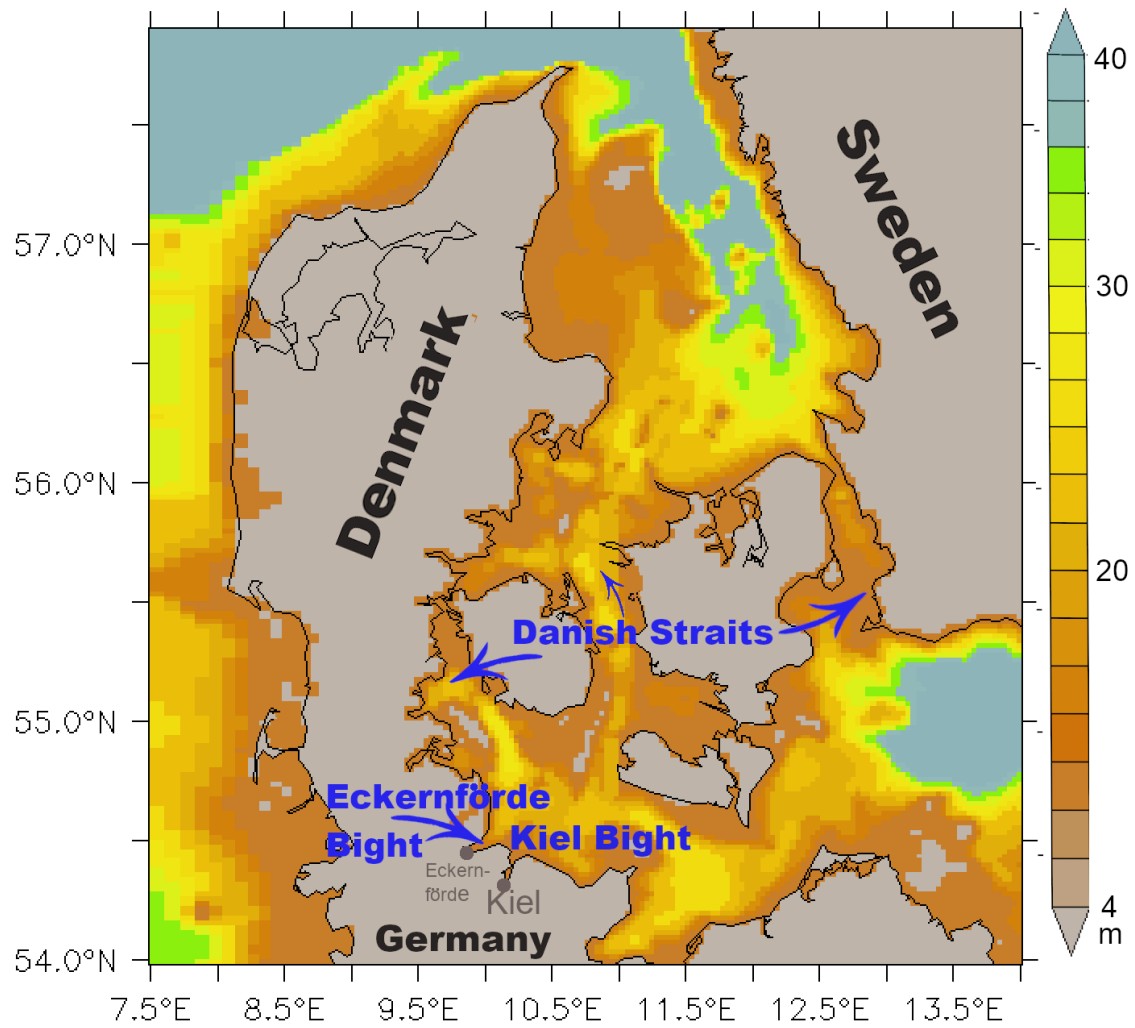

**Figure 1.** Overview map. The colors indicate water depth in m.

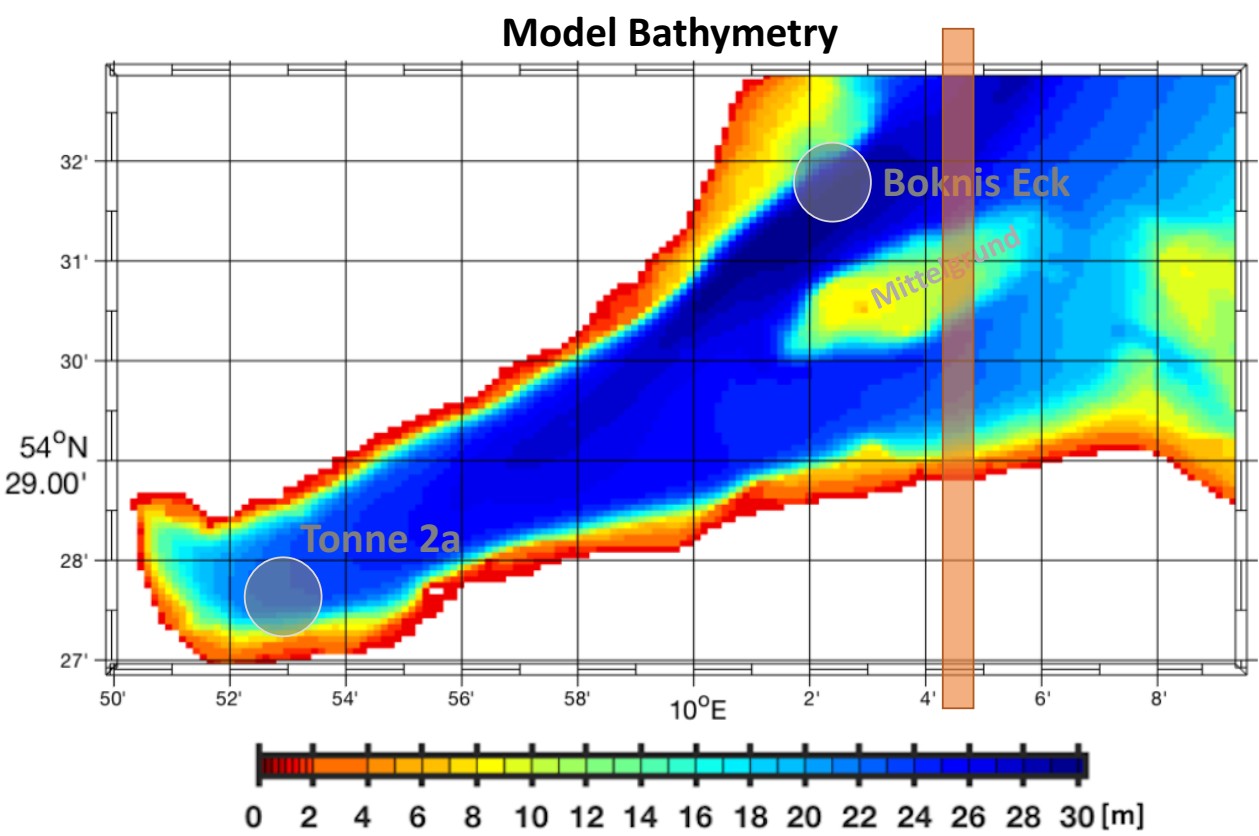

**Figure 2.** Model bathymetry. The horizontal and vertical resolution are 100 m and 1 m, respectively. The northern and eastern boundaries are closed (rigid walls). Sea surface height, temperatures and salinities around the closed boundaries are restored to prescribed values. Grey circles depict the locations of the observational sites at the entrance and deep inside EB. *Mittelgrund* is a shallow. Note that the region east of the orange rectangular is discarded in all following plots because it is essentially determined by our boundary conditions rather than by intrinsic model dynamics.

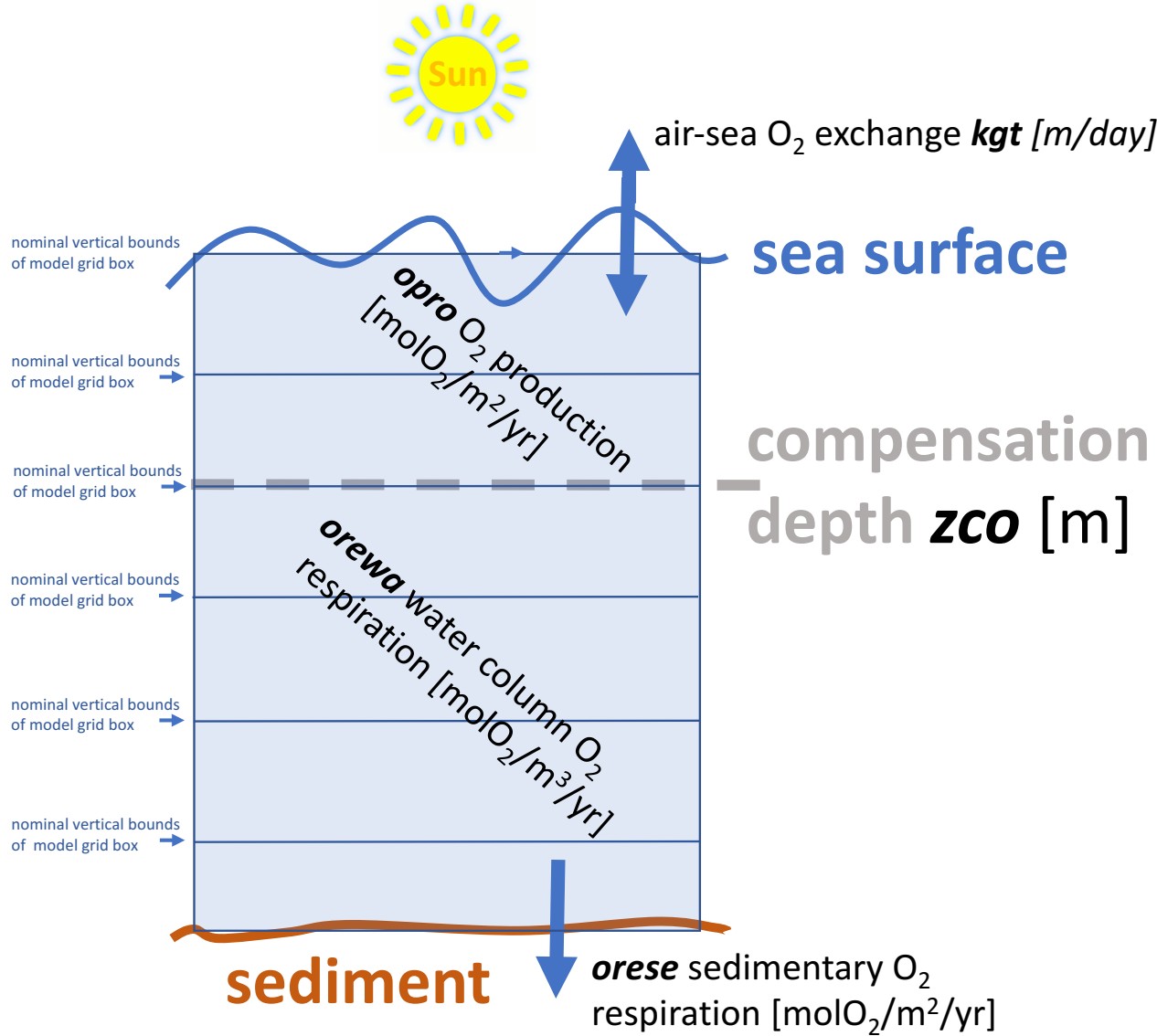

**Figure 3.** Schematic of dissolved oxygen module *EckO₂*. *EckO₂* calculates sinks and sources of oxygen throughout the water column for every grid box. These terms are then passed to the 3-dimensional general ocean circulation that handles the effect of advection and diffusion. The velocity of the air-sea gas exchange is determined by the piston velocity $kgt$. Above the compensation depth $zco$, primary production produces oxygen at a rate prescribed by the model parameter $opro$. Below the compensation depth $zco$, respiration of organic matter consumes dissolved oxygen at a rate prescribed by $orewa$. At the bottom, prescribed oxygen fluxes $orese$ mimic the oxygen consumption of the sediment that is fuelled by the transfer across the water-sediment boundary. Table 2 summarizes respective parameter settings.

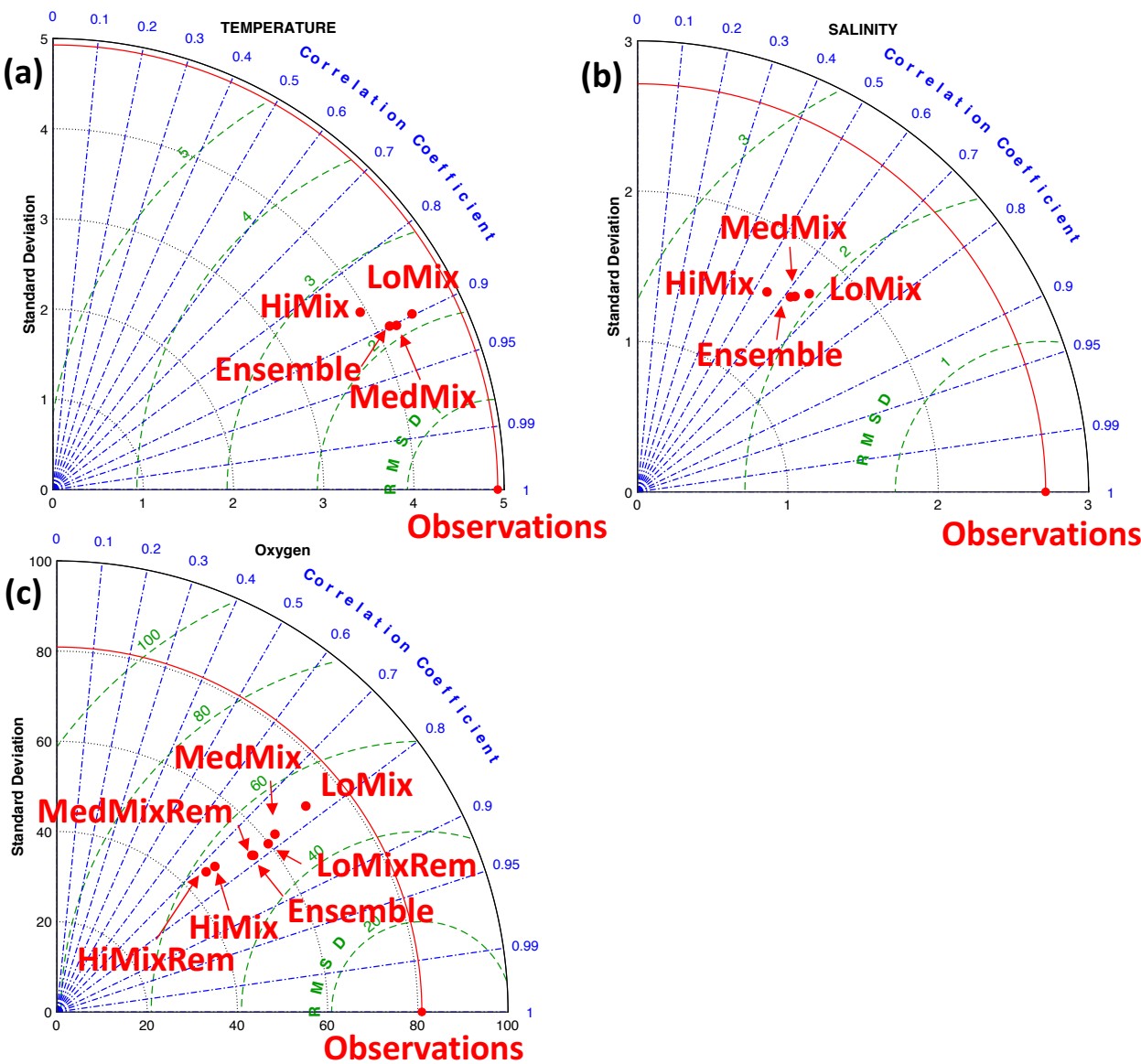

**Figure 4.** Model assessment (Taylor Plots) at Station *Buoy 2a* in the interior of EB (Figure 2). Observational data and model output refer to the 2000 to 2015 period. The simulation tags are defined in Table 2: *LoMix*, *MedMix* and *HiMix* denote the levels of diffusive background mixing. Rem indicates remote effects of biogeochemical sources and sinks of oxygen only (i.e. no local oxygen consumption in EB.

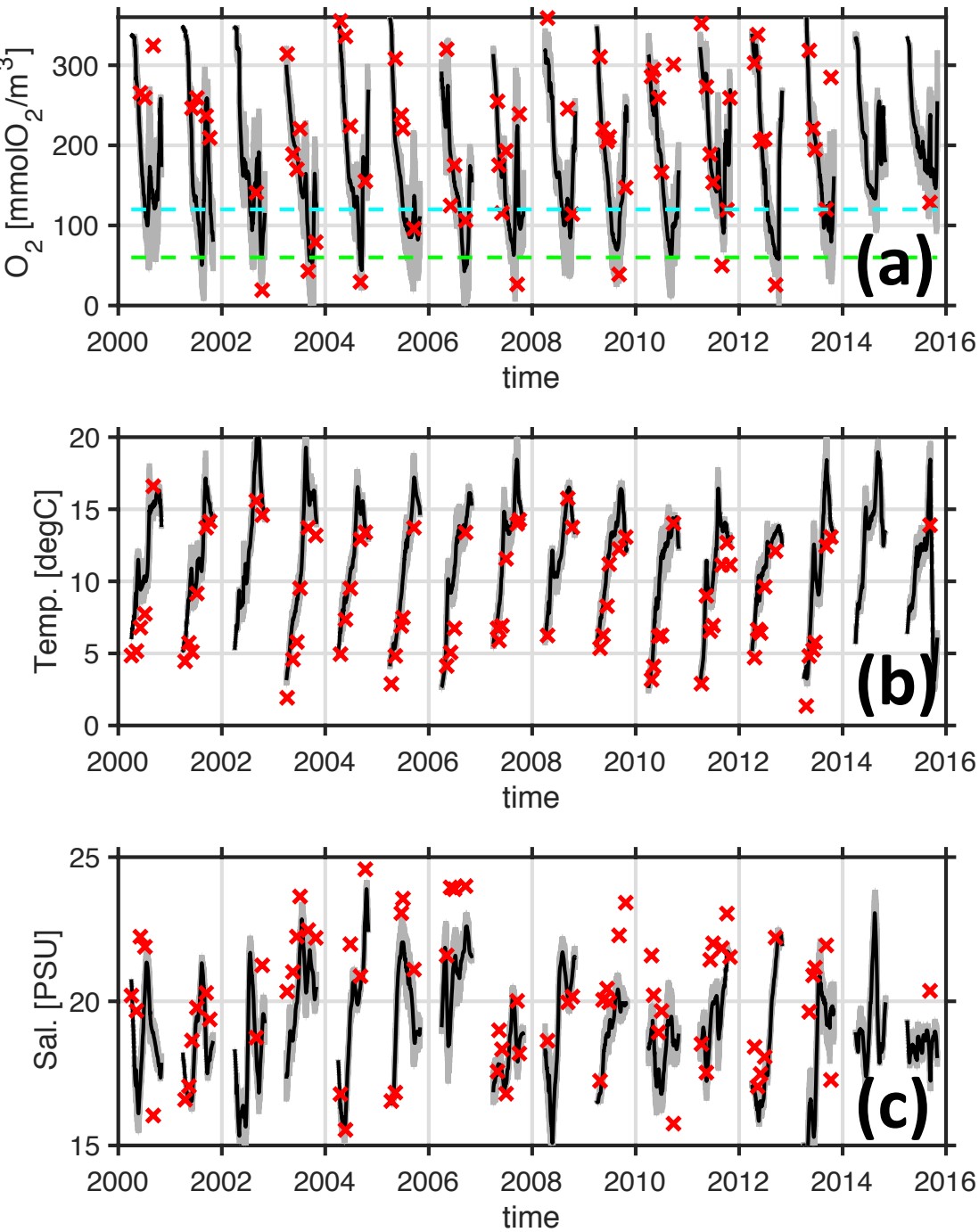

**Figure 5.** Simulated and observed oxygen concentrations at the bottom (20 m depth) of the monitoring station *Buoy 2a*. Panel a, b and c refer to oxygen concentrations, temperature and salinity, respectively. Red crosses denote observations. The black line denotes the ensemble mean of the simulations *MedMix* and *LowMix*. The grey line envelopes the ensembles' extremes at any given time. The horizontal dashed cyan and green lines in panel a show 120 and 60 mmol $O_2$ m$^{-3}$ hypoxia thresholds, respectively.

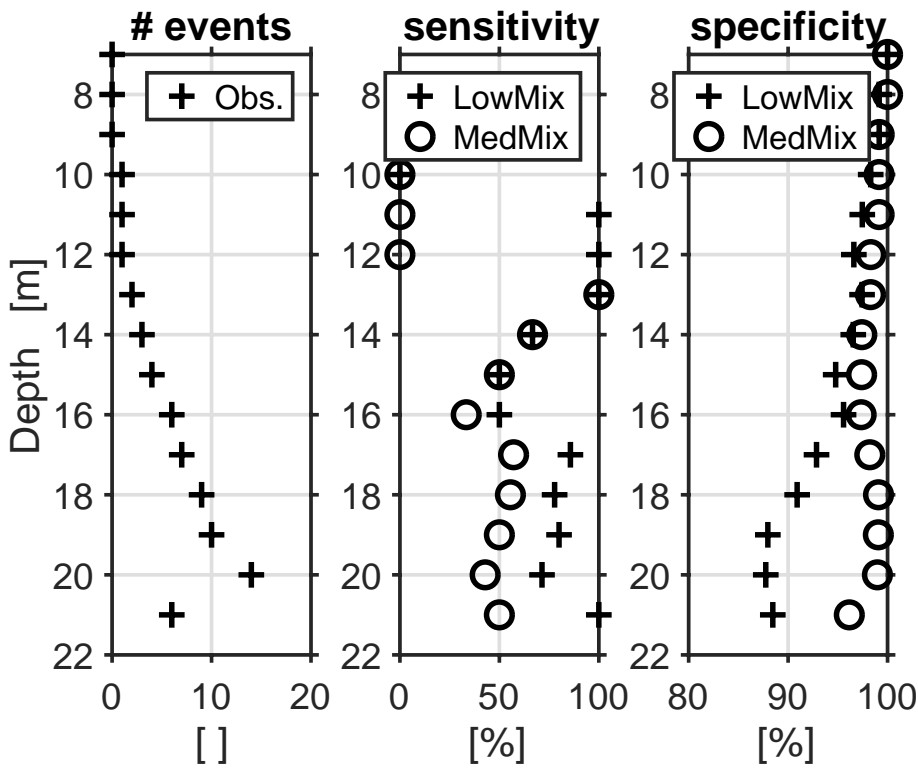

**Figure 6.** Fidelity of hindcasted hypoxic events (oxygen threshold of $120\,\text{mmol}\,O_2\,\text{m}^{-3}$) at Station *Buoy 2a*.

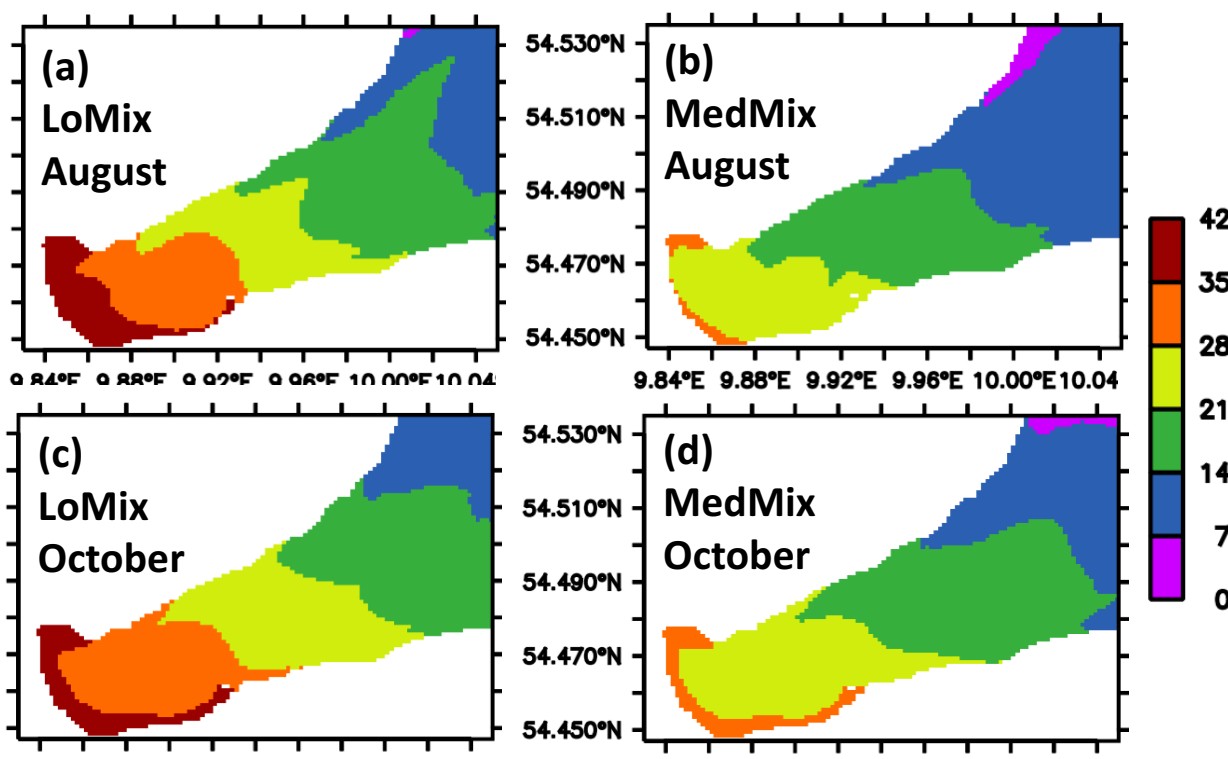

**Figure 7.** Simulated climatological estimate of the residence time of water parcels in EB. The units are days elapsed since the water flushed into the Bight. The estimate refers to the longest residence time found in local water columns. Panels (a) and (b) refer to August calculated by the simulations *LowMix* and *HiMix*, respectively. Panels (c) and (d) refer to October calculated by the simulations *LowMix* and *HiMix*, respectively. Note that the model domain extends beyond the eastern boundary shown here (see also Figure 2).

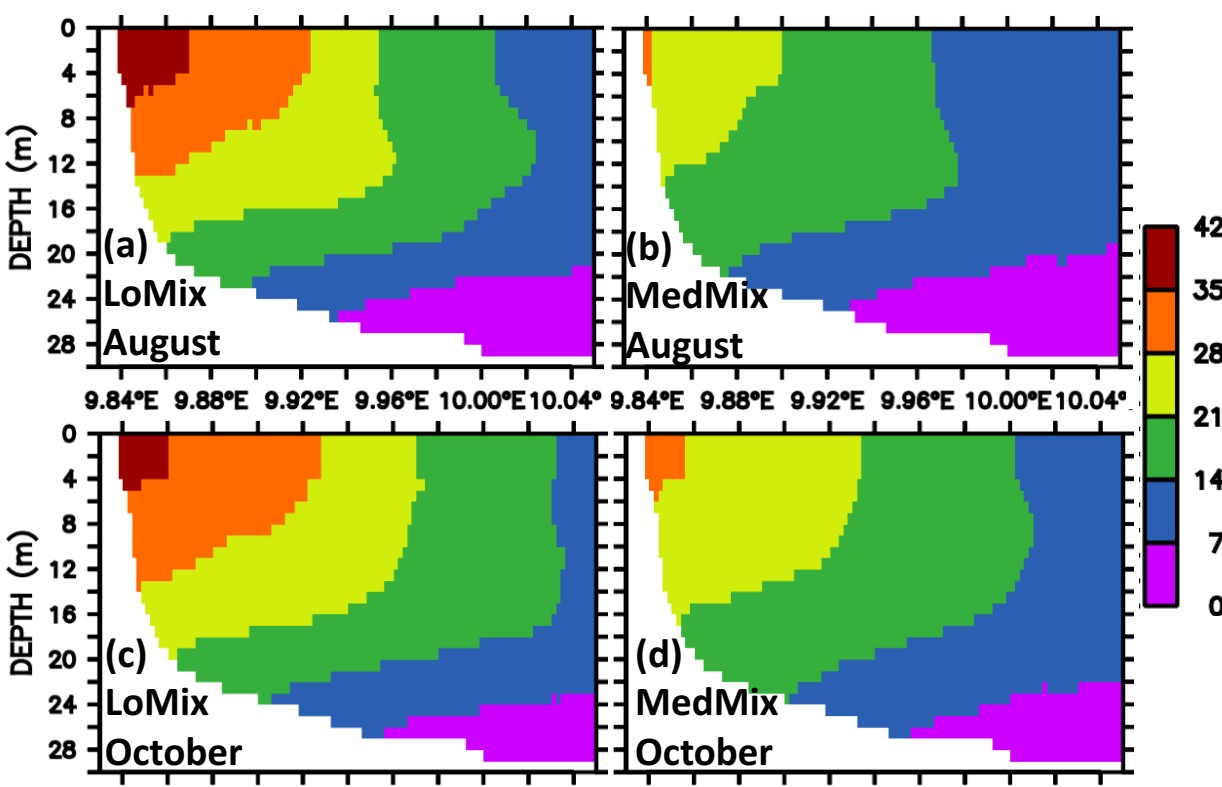

**Figure 8.** Simulated climatological estimate of the residence times of water parcels in EB. The units are days elapsed since the water flushed into the Bight. Shown are sections along EB. Panels (a) and (b) refer to August calculated by the simulations *LowMix* and *HiMix*, respectively. Panels (c) and (d) refer to October calculated by the simulations *LowMix* and *HiMix*, respectively. Note that the model domain extends beyond the eastern boundary shown here (see also Figure 2).

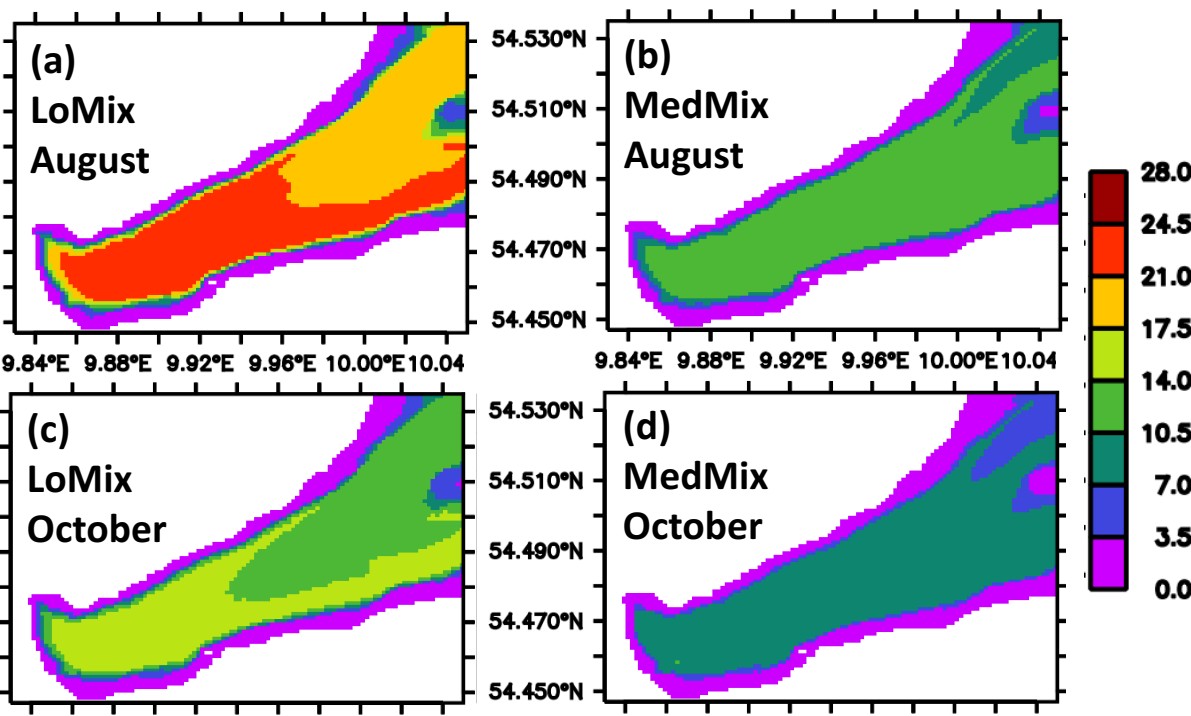

**Figure 9.** Simulated climatological estimate of local ventilation. The color shading denotes the time elapsed (age) since bottom water has been in contact with the atmosphere in units days. Panels (a) and (b) refer to August calculated by the simulations *LowMix* and *HiMix*, respectively. Panels (c) and (d) refer to October calculated by the simulations *LowMix* and *HiMix*, respectively. Note that the model domain extends beyond the eastern boundary shown here (see also Figure 2).

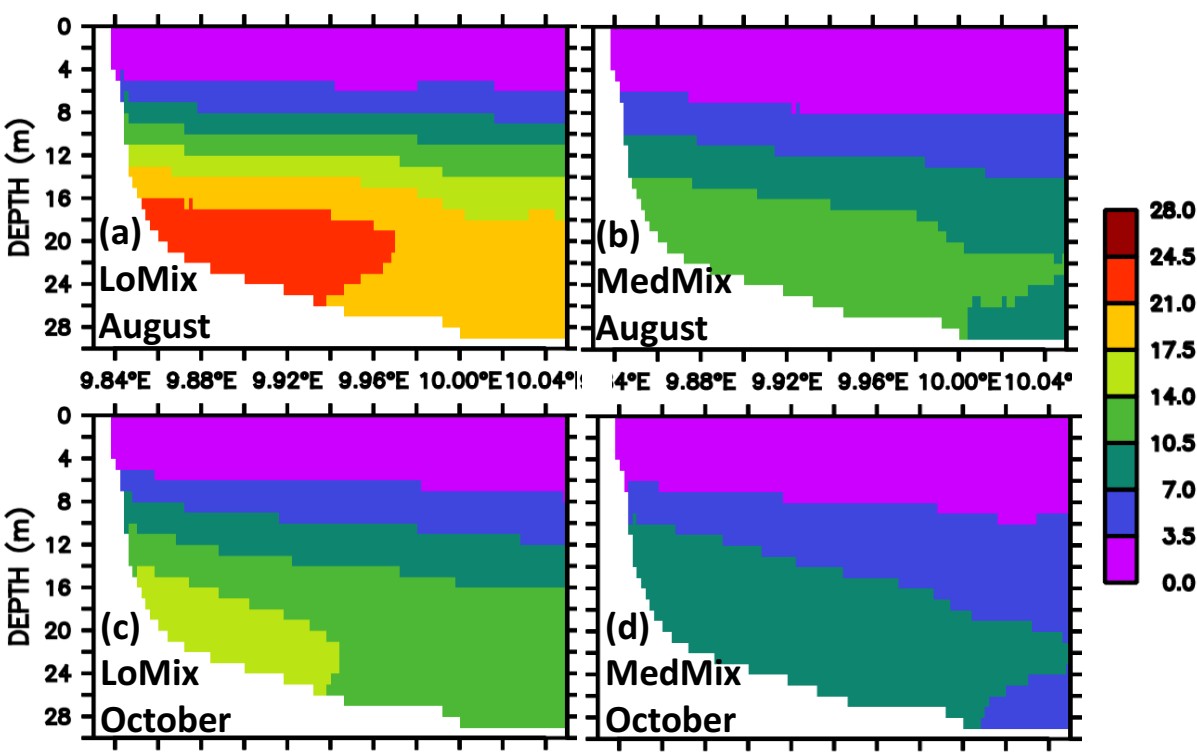

**Figure 10.** Simulated climatological estimate of local ventilation. The color shading denotes the time elapsed (age) since water parcels have been in contact with the atmosphere in units days. Shown are sections along EB. Panels (a) and (b) refer to August calculated by the simulations *LowMix* and *HiMix*, respectively. Panels (c) and (d) refer to October calculated by the simulations *LowMix* and *HiMix*, respectively. Note that the model domain extends beyond the eastern boundary shown here (see also Figure 2).

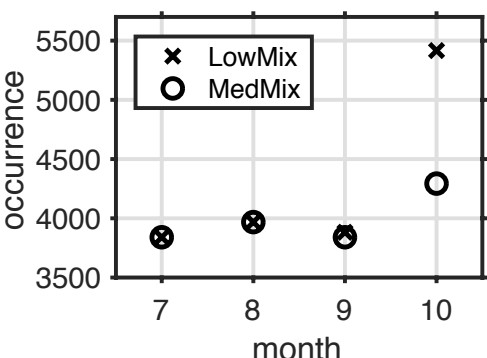

**Figure 11.** Simulated climatological (2000 - 2015) occurrence of hypoxia at the monitoring station *Buoy 2a*. Occurrence refers to the sum of suboxic (i.e., $<120\,\mathrm{mmol\,O_2\,m^{-3}}$) model grid boxes, identified in climatological daily model output. From November to June no suboxic conditions were absent.

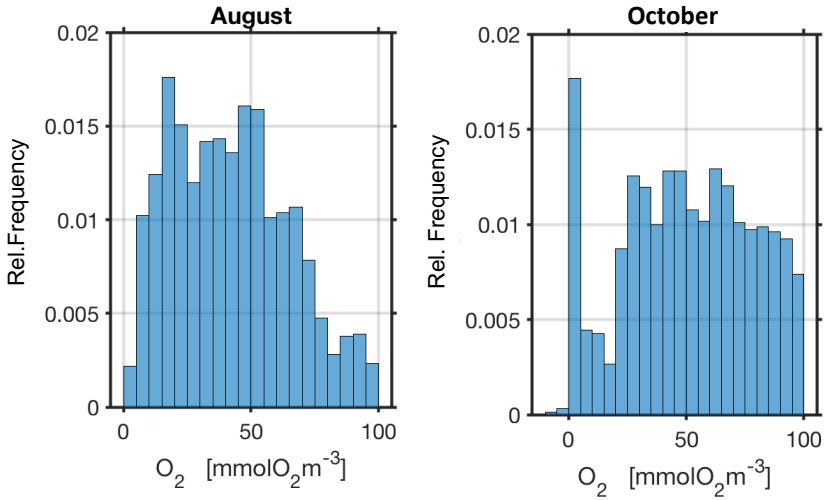

**Figure 12.** Histogram of observed climatological bottom oxygen concentrations at Boknis Eck (capped at $100\,\mathrm{mmol\,O_2\,m^{-3}}$).

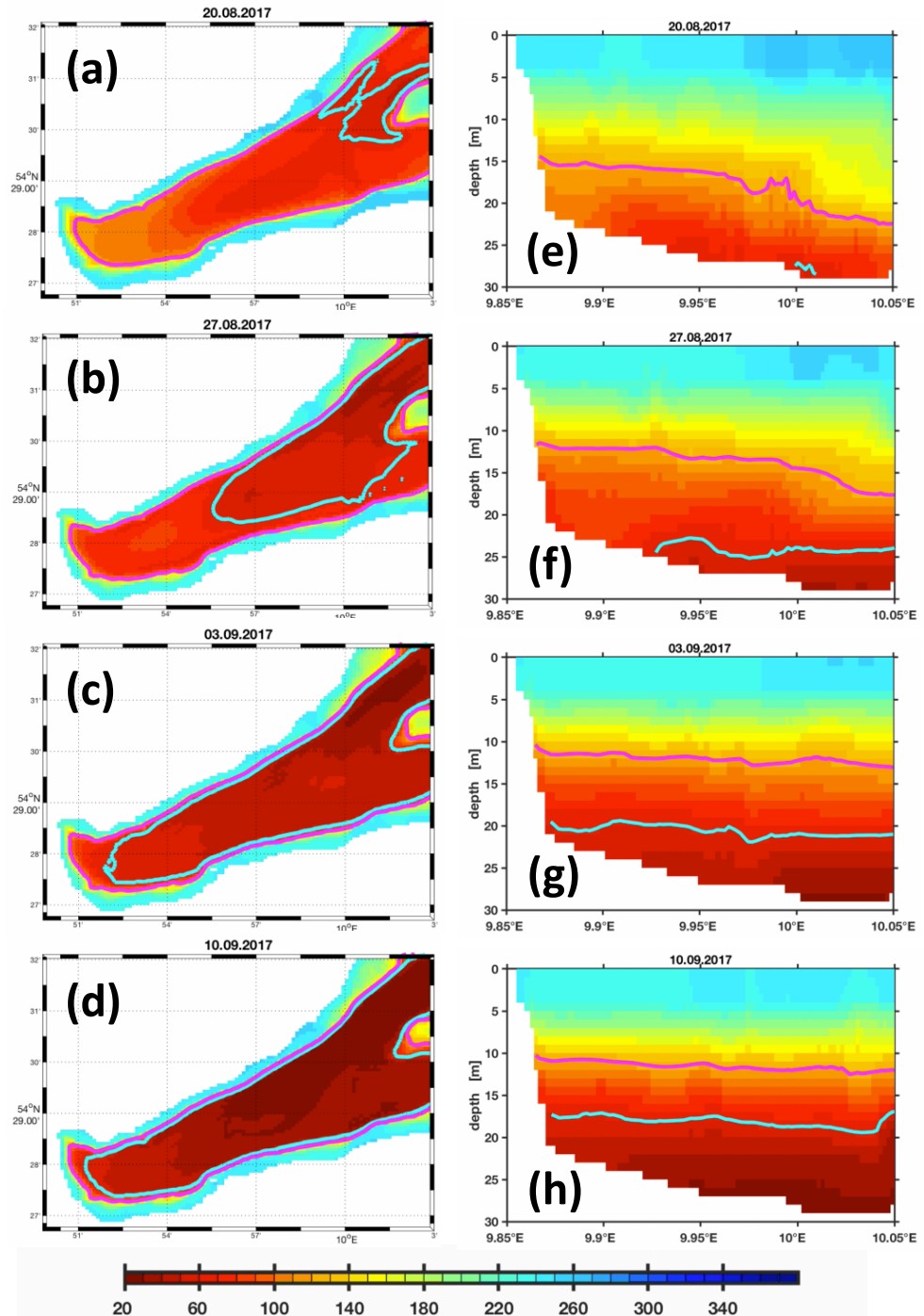

**Figure 13.** Simulation (*LoMix*) of the 2017 hypoxic event. The colors refer to oxygen concentrations in $\mathrm{mmol\,O_2\,m^{-3}}$. The contours in cyan and magenta show the 60 and $120\,\mathrm{mmol\,O_2\,m^{-3}}$ isolines. The left column (Figures a to d) show oxygen concentrations on the sea floor. The right column (Figure e to h) shows a section through the Bight with the city of Eckernförde to the left and the entrance to the Bight to the right. (Corresponding animations featuring daily resolution named *LowMix_O2_Bottom_2015.m4v* and *LowMix_O2_zonal_2017.m4v* are archived at https://doi.org/10.5281/zenodo.4271940.) Note that the model domain extends beyond the eastern boundary shown here (see also Figure 2).

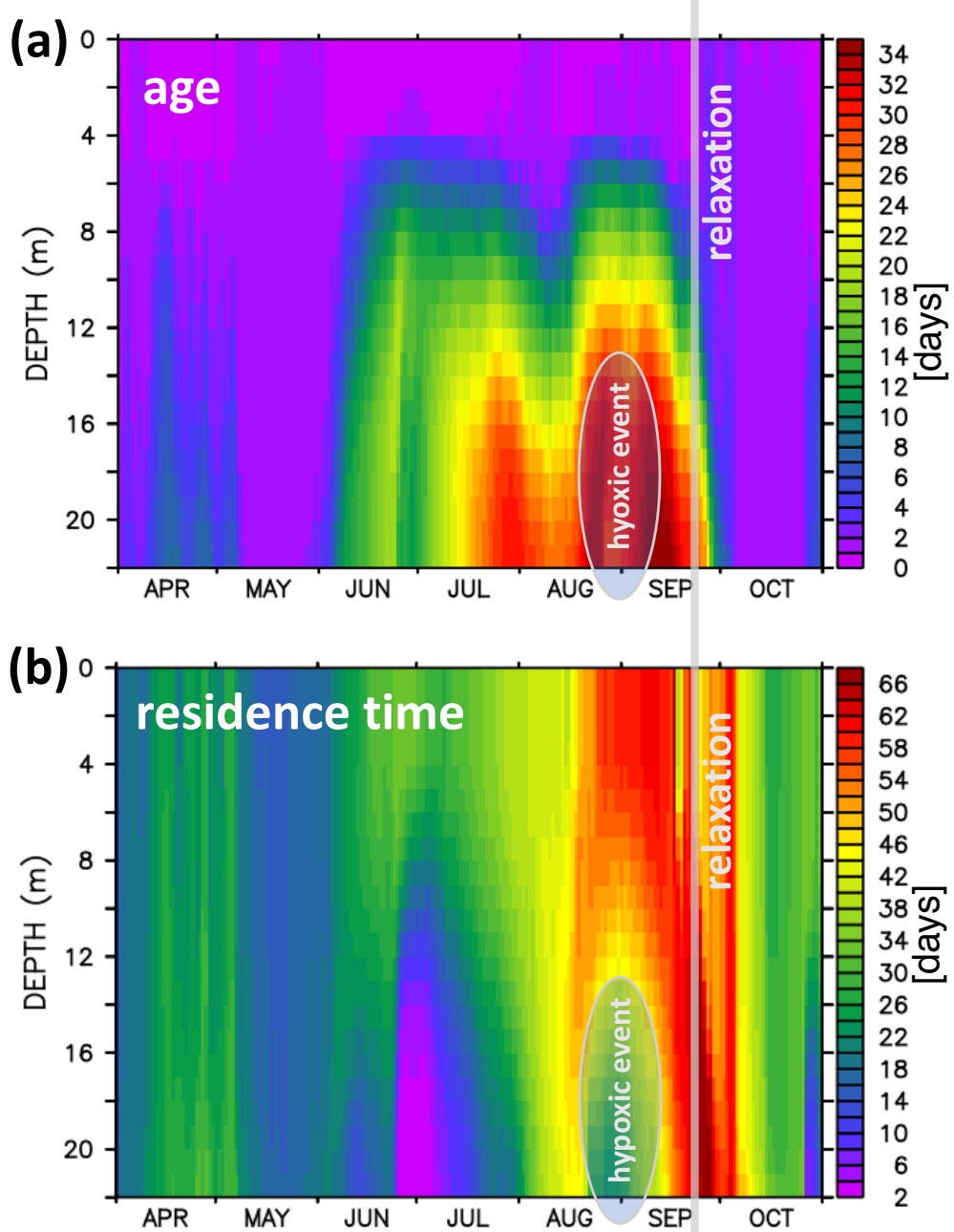

**Figure 14.** Hovmoeller Diagrams of simulated water age and residence time at the monitoring station *Buoy 2a* (panel a and b, respectively). The oval marking in August - September highlights the 2017 hypoxic event. The vertical gray line marks the start of the relaxation phase, ending the hypoxic event.

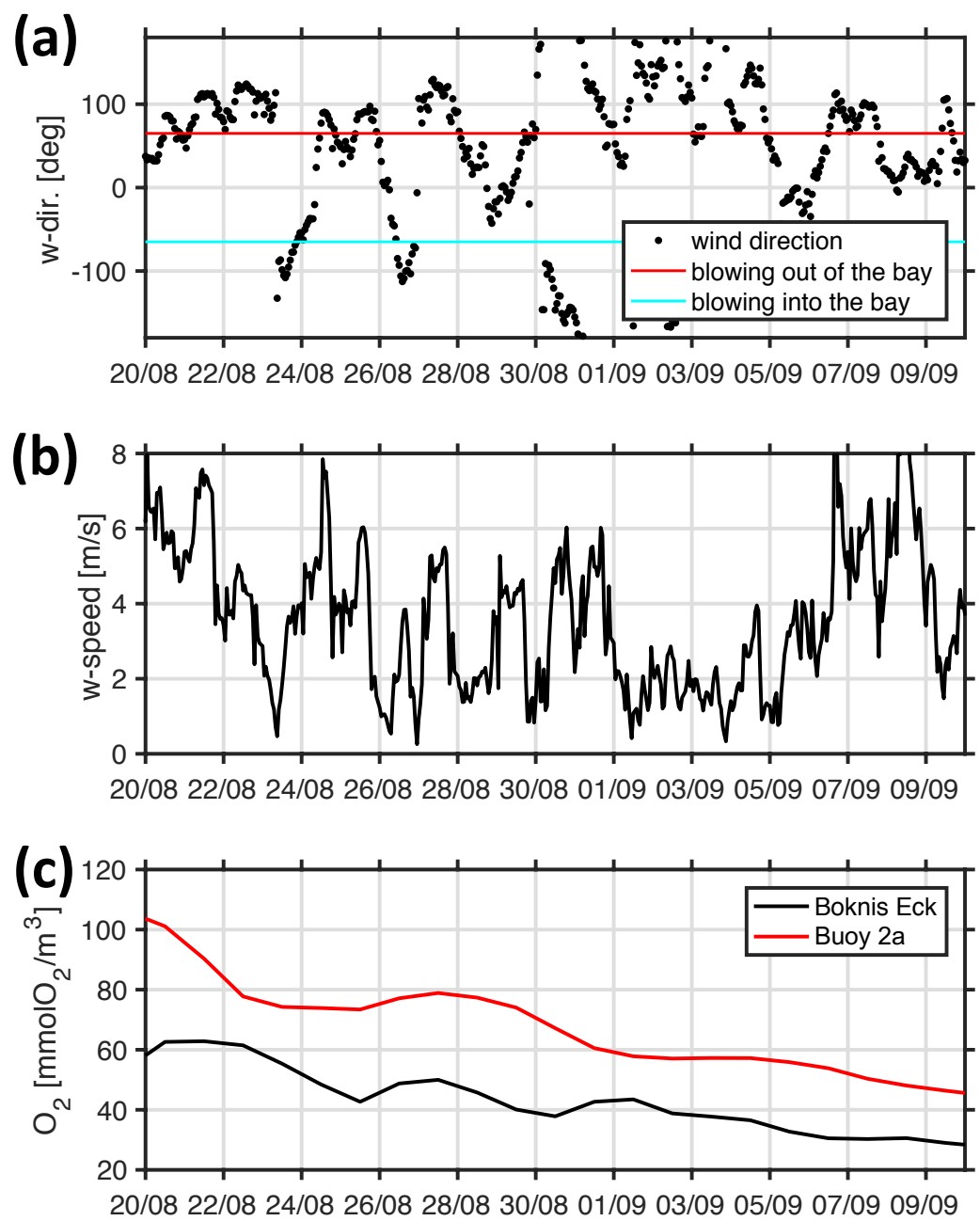

**Figure 15.** Simulated temporal evolution of wind direction, wind speed and bottom oxygen concentrations during the buildup of the 2017 hypoxic event. Panel a, b and c show wind direction, wind speed and bottom oxygen concentrations at the entrance (Station *Boknis Eck*) and deep inside EB (Station *Buoy 2a*).

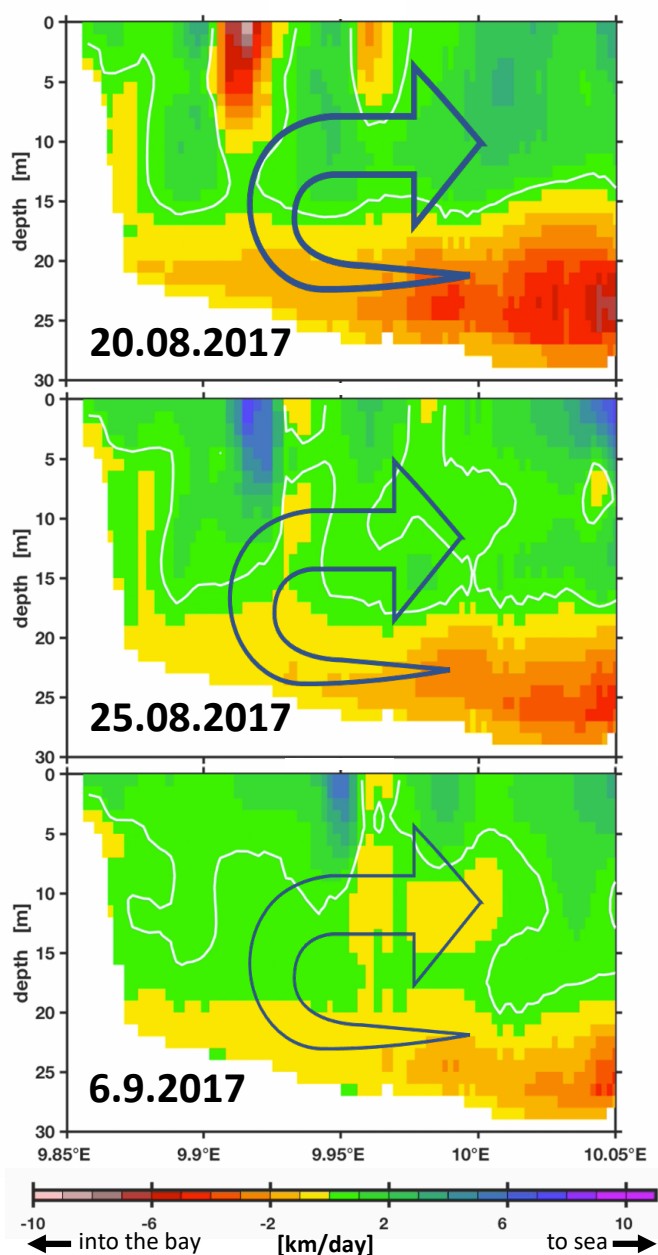

**Figure 16.** Simulated, daily mean zonal currents during the buildup of the 2017 hypoxic event shown in Figures 13, 14, and 15. Green to blue colors characterize flows to the east (towards the KB). Yellow to red colors indicate flows to the west (into EB). The unit is km per day. The depicted section has an extension of $\approx 13\,\mathrm{km}$. Note that the model domain extends beyond the eastern boundary shown here (see also Figure 2).

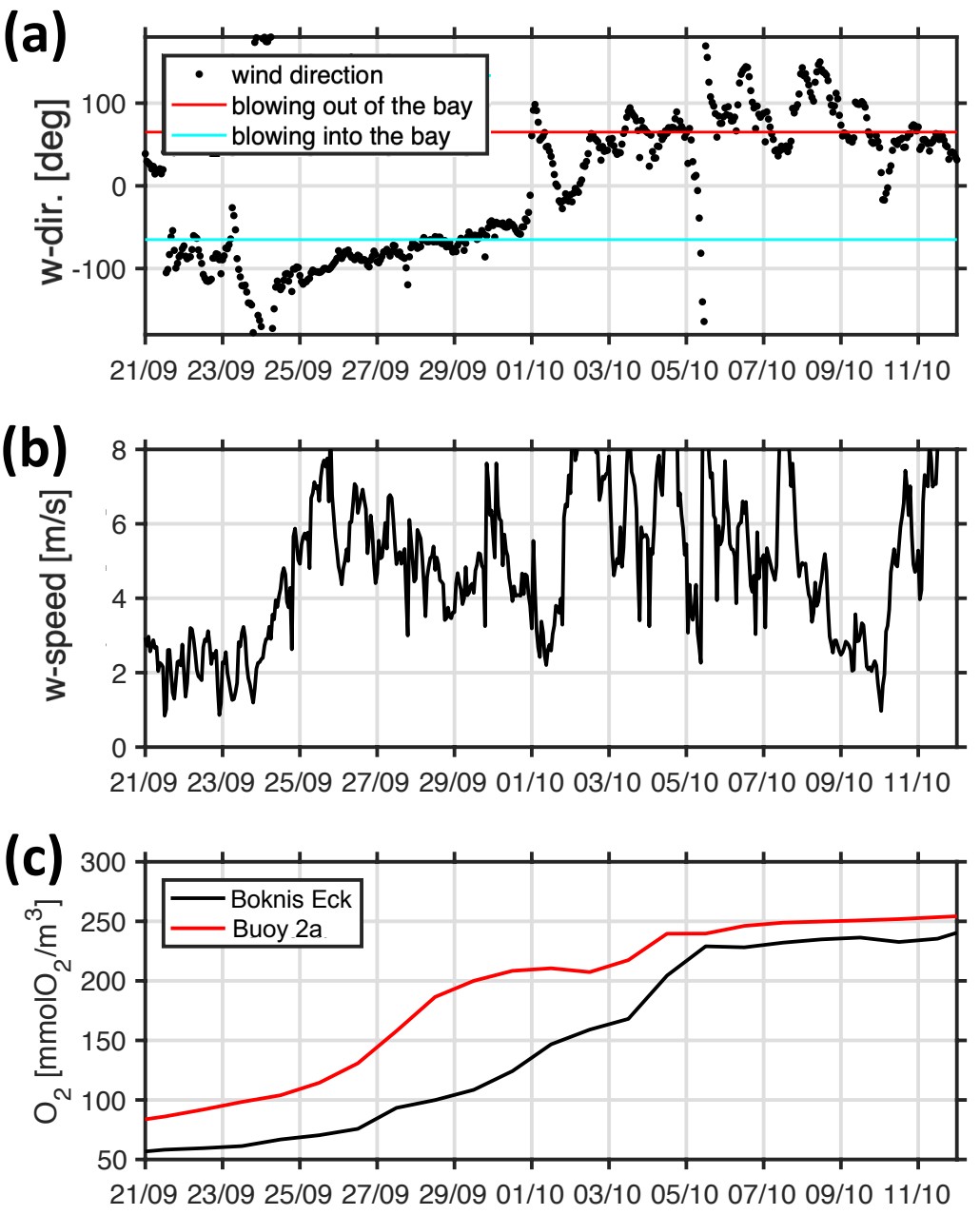

**Figure 17.** Simulated temporal evolution of wind direction, wind speed and bottom oxygen concentrations during the relaxation phase that terminates the 2017 hypoxic event. Panel a, b and c show wind direction, wind speed and bottom oxygen concentrations at the entrance (Station *Boknis Eck*) and deep inside EB (Station *Buoy 2a*).

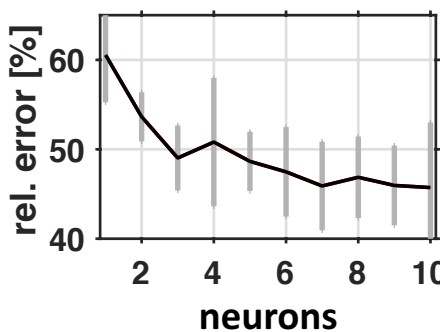

**Figure 18.** ANN error relative to naive persistency forecast versus the number of neurons in the hidden layer. The black line features the best ANN parameter setting found within an ensemble of 30 optimizations for each of the number of neurons tested. The grey bars denote the ensemble's standard deviations.

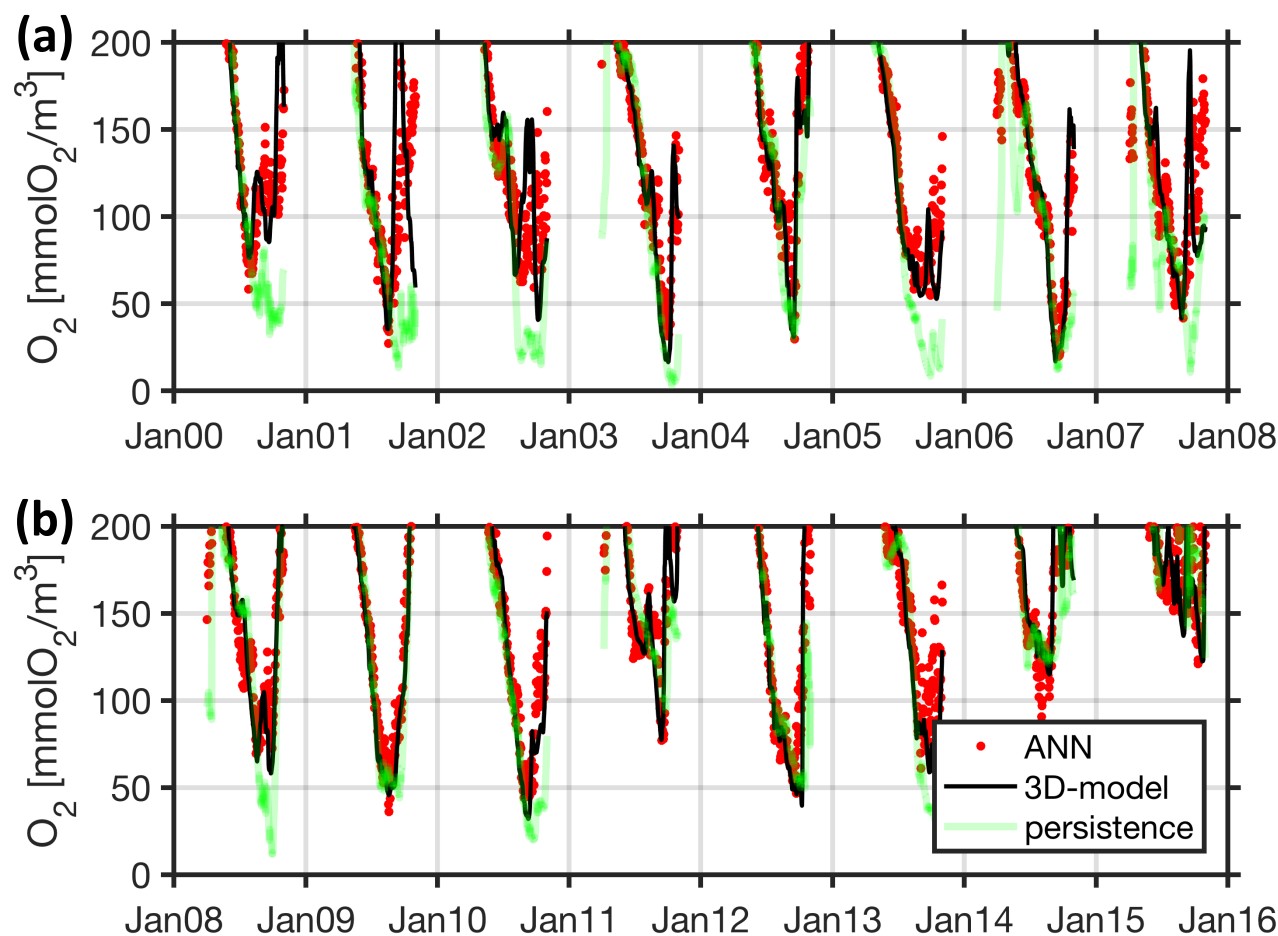

**Figure 19.** Walk-forward performance of ANN based on training and testing data (corresponding to 80% and 20% of the data shown here). The black line shows bottom oxygen concentrations at Station *Buoy 2a* as simulated with the full-fledged and computationally expensive 3-D coupled ocean-circulation biogeochemical model. Each of the red dots denotes a respective biweekly walk-forward (computationally cheap) ANN forecast utilizing surface and bottom temperatures at Station *Boknis Eck* only. For comparison, the green line features a naive biweekly persistency forecast based on bottom oxygen concentrations at Station *Boknis Eck*.

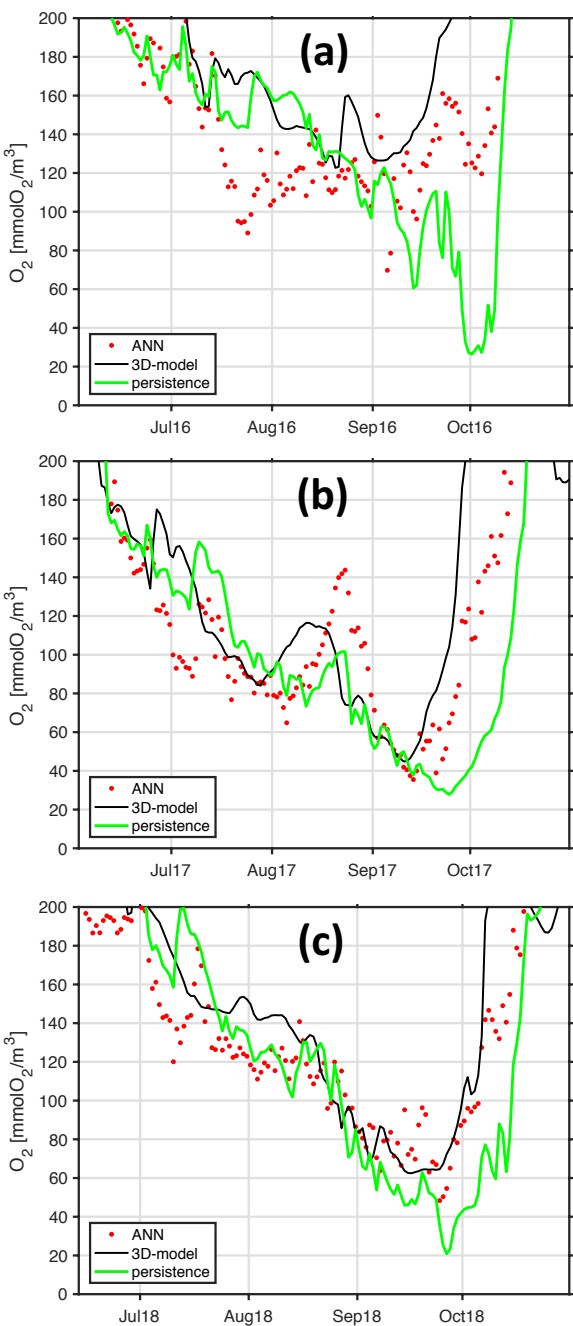

**Figure 20.** Walk-forward validation (generalization) of ANN. The panels a, b, and c refer to year 2016, 2017, 2018. The black line shows bottom oxygen concentrations at the monitoring station *Buoy 2a* as simulated with the full-fledged and computationally expensive 3-D coupled ocean-circulation biogeochemical model. Each of the red dots denotes a respective biweekly walk-forward (computationally cheap) ANN forecast utilizing surface and bottom temperatures at Station *Boknis Eck* only. The green line features a naive biweekly persistency forecast based on bottom oxygen concentrations Station *Boknis Eck* for comparison.