# Peer review of "Retracing Hypoxia in Eckernförde Bight (Baltic Sea)"

_Biogeosciences, 2021_

## Author Response (AR1)

Kiel, May 26th, 2021

Dear Editorial Team,

thank you very much for your interest in our manuscript. Thank you also very much for (potentially) adding us to the EB Special Issue.

We revised our manuscript substantially following up the reviewers' apparent confusions and very constructive suggestions. Our major aim during revision was to (1) make the manuscript more appealing to a wider (non-Kiel-based) audience and get away from a report-style presentation following Reviewer #1 and (2) resolve the confusion of Reviewer #2 which we think, was triggered by us doing a bad job of making clear where our model domain actually ends.

We hope that the substantially revised manuscript now meets the standard for publication in Biogeosciences.

In any case: thank you for your work and time!

Yours Sincerely,

the authors

PS: Attached please find point-by-point responses to the reviewers' comments. **The line numbers refer to the pdf-document with highlighted changes.**

**Reviewer #1**

A: Thank you for your instructive questions which helped to modify our storyline. We hope we found a new focus such that the manuscript is now more appealing to the wider audience addressed by Biogeosciences. **Please note that, in the following, all page and line numbers refer to the "changes highlighted" version of the revised manuscript**.

**R#1: 1. The key findings of the manuscript are that O2 dynamics in Eckernförde Bight are determined by the inflow of water from the adjacent Kiel Bight and mixing processes within Eckernförde Bight. Was this unknown prior to this study?**

A: The new tweak here is the importance of wind-induced downwelling which ends the hypoxic season as soon as a weakened stratification allows for it. We added a respective discussion starting pg. 15, 459 and also totally overhauled the introduction in order to set the reader on the right track.

**R#1: Physical processes drive most of the observed oxygen variability in coastal systems, so were these findings unexpected?**

A: Deconvolution of forcing effects of hypoxia is typically very difficult (Naqvi et al. 2010). So, one may argue that the unexpected finding is that disentangling biogeochemical effects from physical apparently was possible. Also, maybe somewhat unexpected, was the clear effect of stratification in gatekeeping hypoxia (which relates hypoxia with climate change). We made this clearer in the revised version of the manuscript (starting pg. 15, ln. 485).

**R#1: 2. There are 3 Tables and 23 Figures!  I think some editorial evaluation is needed about whether to place some in a Supplement or prioritize the Figures for publication.  Inclusion of all the figures causes the manuscript to read more like a technical report rather than a publication.  This is not a bad thing, but it did make me wonder why the authors chose Biogeosciences journal**

A: We deleted 3 Figures in the revised version of the manuscript. In addition, we made substantial changes in the introduction and conclusions in order to make the paper more appealing to a wider audience.

**R#2: 3. Does this work have any wider ramifications for other coastal sites?  The Discussion/Conclusion focus exclusively on Eckernförde Bight with no mention of using the simulations to improve predictions at other locations.  Again, I think if the authors publish in a journal with wide readership then a broader context is needed.**

A: Our selling points that may have wider ramifications for other coastal sites are: (1) De-convolution of processes causing hypoxia can be achieved by a combination of coupled oceancirculation biogeochemical modelling and AI. (2) Showcased importance of wind induced downwelling. This may be surprising (or overlooked at other sites) because, typically, the focus is on its upwelling counterpart which fuels phytoplankton at the sun-lit surface with nutrients thereby driving export of organic matter to depth where it is remineralized by heterotrophic bacteria consuming oxygen. We rewrote part of the conclusions (starting pg. 16, ln. 509 and starting pg. 17, ln.17).

**R#1: 4. Please can the authors improve their communication about whether or not the simulations sufficiently predicted the oxygen dynamics. I realize this is shown in Figure 7 and 8, and discussed in the text on Lines 177 and 358, but it is unclear to me whether the correlation coefficient obtained for the simulations is deemed to be successful or needing further work.**

A: Please note, that we changed the overall storyline to make the manuscript more appealing for a wider audience ( - triggered by your other comments). We changed from "we predict o2 for a stakeholder" to "we set out to quantify drivers of the dynamics of hypoxia and found wind-induced downwelling gatekept by stratification which may be important elsewhere also". Basically, we made substantial changes to the Introduction, Discussion and Conclusion such that we feel a discussion regarding the sufficiency of fidelity for the stakeholder is no longer required. Even so, the information relevant for the stakeholder is still there: Figure 6 summarizes the fidelity of our model-based forecasts of hypoxic conditions (discussed starting pg. 9, ln. 261). Our best model features a sensitivity better than 75% and a specifity of around 90% at depth (i.e. below 16m). The discussion with the stakeholder regarding sufficiency for specific purposes are ongoing ...

**Reviewer #2**

A: Thank you for your time and careful examination of our simulated oxygen concentrations. We want to apologize because we think that our choice of the eastern x-axes limit of our 2-D plots which sliced through a shallow - rather than corresponding to the eastern boundary condition - put you on the wrong track. We hope that, now that we make clear that the "eastern oxygen anomaly" is a consequence of a shallow rather than proof of a retarded boundary condition, we regain your trust in our modelling approach. **Please note that, in the following, all page and line numbers refer to the "changes highlighted" version of the revised manuscript**.

**R#2: 1. The manuscript in its current state lacks literature on general circulation in the EB or the Baltic Sea to support later findings that low oxygen within this bay is imported and not due to local process. I'm assuming there is literature on the physics close to EB to give the reader a general idea to draw conclusions?**

A: We added recent literature on hypoxia at the entrance of EB including a discussion on drivers of an apparently increasing number of events (Hoppe et al. 2013, Lennartz et al. 2014 ) and of wind-induced upwelling (Karstensen et al. 2014) (pg. 4, ln. 98; pg. 14, ln. 439; pg. 16, ln. 510). As for the circulation in EB we, unfortunately, did not find references. Because we think that your comment reverberates the major issue raised by Reviewer #1 - that is that we should become more appealing for a wider (non-Kiel-based) audience - we hope that the substantially reworked introduction does now give the reader more information to draw conclusion (and, thus, make it more appealing).

**R#2: 2. Due to the rigid walls in the northern and eastern boundaries the model acts like a "tank" and to a certain extent is not suitable for resolving remote processes. While restoration can be effective in constraining the model to prescribed values it does not replace the effectiveness of open boundary conditions.**

A: We agree that boundary conditions are a key element in regional modelling. We added a respective discussion on why we use rigid walls and how this may affect our results and conclusions on: pg. 6, ln. 171 and pg. 16, ln. 500. Basically we make the point that our major results, the high importance of import/export of hypoxia, is robust towards changes to the boundary conditions because it is hard to argue that open boundary conditions would allow for less exchange between KB and EB.

**R#2: 3. The model is able to capture temperature, salinity and oxygen concentrations because its not allowed to drift freely due to the restoration. Hence the good representation from the MedMix and LoMix experiments in the Taylor diagrams. These two experiments may be representing less "aggressive" diffusivity which is also not countered by inflow as compared to the HiMix hence the good agreement. So in its current state the model is mostly suitable for resolving vertical processes as drawn from conclusions of the evaluation.**

A: We are sorry for the confusion. There is no restoring inside EB. We state this (now) explicitly on pg. 6, ln. 180.

**R#2: 4. The higher oxygen concentrations between 30–31°N and beyond 10°E in Figure 16 suggest that there is an almost permanent feature at the boundary possibly due to the restoration and potentially weak boundary conditions which may not be strong enough to push this into or out of EB. Hence the comment that this model is currently suitable for investigating vertical processes. A hovmuller plot of dissolved oxygen in addition to the water age and residence time in Figure 17 could paint a better picture of how these factors are related.**

A: We are sorry for the confusion. Figure 16 shows only the "Eckernfoerde Bay" part of the model domain. The actual domain is larger as indicated in the Model Bathymetry Figure. Hence the "almost permanent feature" is not at the boundary. It is part of the shallow Mittelgrund which, because it raises up to better ventilated surface waters features almost permanently higher oxygen concentrations than its surroundings.  The revised version of the manuscript (now) tags the shallow Mittelgrund and shows where the plots end in relation to the model boundaries in Figure 2. Further, wherever appropriate, we state in all figure captions that the actual model boundaries are not corresponding to the limits of the plotting axes shown (e.g. in Figure 13).

**R#2: 5. Machine learning techniques are generally good in forecasting if fed enough data. The reason why there is discrepancies and between ANNs and the model is because wind may not be driving the flow in the model in EB, again, due the rigid walls and weak boundary conditions. The ANNs are able to perform well with just temperature and not wind, and this should be investigated further as its concerning. It may imply that the oxygen is consumed within EB but this can only be resolved if the boundary conditions issue is resolved.**

A: We are sorry for the misunderstanding. The ANN is applied and trained using only model output and not observed values. We repeated this information now on pg. 13, ln. 406. In our model, we know (and prove, e.g. in Figure 20) that there is a strong effect of the wind. This effect is, however, not picked up by our ANN which is trained with the model output. Still, the ANN performed surprisingly well and suggested that the seasonal cycle of oxygen concentrations is surprisingly predictable using information on temperature and stratification only (especially given the limited amount of training data from our model simulations). We extended the respective explanation on pg. 14, ln. 424 - ln. 435.

**R#2: 6.The authors can use the model in its current state to investigate vertical processes or address the boundary conditions issue which is evidently persistent in the results.**

We disagree. We argue that there is no boundary conditions issue but that the reviewer has misinterpreted the effect of the shallow Mittelgrund on bottom oxygen concentrations: Since Mittelgrund is closer to the well-ventilated surface, there, bottom oxygen concentrations are higher than ambient ones corresponding to greater depths. We realized that this misinterpretation was triggered by our unfortunate or uncommented choice of plot-axes-limits. The revised version of the manuscript now makes clear that many plots slice through Mittelgrund (tags in Figure 2; last sentence of captions to 2-D figures). We apologize for the confusion caused by our presentation.

---

## Author Response (AR2)

Dear editorial team,

thank you for your work!

Following the editor's suggestion, we replaced "men" with "humans" in the first sentence of the introduction.

In addition, we fixed the following very minor issues:

- one broken referencing to a picture reading "??" instead of the correct figure number
- "my suffice" ==> may suffice
- adjusted acknowledgments to symbols that were used in Figure 1 of the first version of the manuscript in the acknowledgment section (the original Figure 1 was deleted during revision but is now reused as what you term "key figure")
- added an acknowledgment to our anonymous reviewers and the editor in the acknowledgment section

Kind regards,

Heiner on behalf of the authors.